# Maximal Gauge Symmetry in Transformer Architectures

## Abstract

Modern Transformers possess redundant parameter symmetries that leave their function unchanged. We establish the complete gauge group structure for the canonical Transformer family, which encompasses standard architectures including GPT-2, BERT, LLaMA, and Qwen. For canonical Transformers with standard multi-head attention, we prove global maximality: the gauge group equals exactly $G_{\max} = ((\mathrm{GL}(d_k))^h \times (\mathrm{GL}(d_v))^h) \rtimes S_h$ on the generic stratum where projection matrices have full column rank and head-wise attention controllability holds. For architectures with rotary position embeddings (RoPE) or relative encodings, as used in LLaMA and Qwen, the gauge group becomes $G_{\mathrm{RoPE}} = ((\mathcal{C}_{\mathrm{RoPE}})^h \times (\mathrm{GL}(d_v))^h) \rtimes S_h$ where $\mathcal{C}_{\mathrm{RoPE}}$ is the commutant of the position-dependent rotations—typically reducing to $(\mathrm{GL}(1,\mathbb{C}))^{d_k/2}$ for standard RoPE implementations. We prove maximality through three key results: characterizing the Lie algebra of infinitesimal symmetries as $\mathfrak{g}_{\max} = \bigoplus_{i=1}^{h} \mathfrak{gl}(d_k) \oplus \bigoplus_{i=1}^{h} \mathfrak{gl}(d_v)$ for canonical models, establishing that attention weights must be preserved up to head permutation under gauge equivalence, and demonstrating that query-key and value-output transformations necessarily factorize independently. These gauge symmetries persist through LayerNorm and extend to complete architectures, with the full model gauge group being $G_{\mathrm{Model}} = \prod_{l=1}^{L} G_{\mathrm{Layer}}^{(l)}$. Our characterization reveals over 1.1 million redundant dimensions in a 110M parameter Transformer Base model. Experiments confirm gauge transformations preserve outputs to within $24\varepsilon_{\mathrm{mach}}$ relative error across diverse architectures while transformations outside $G_{\max}$ produce $O(1)$ changes, empirically supporting maximality. For grouped/multi-query attention (GQA/MQA), we show the admissible query–key and value transforms are tied per K/V group, yielding a reduced symmetry $G_{\mathrm{share}} = ((\mathrm{GL}(d_k))^g \times (\mathrm{GL}(d_v))^g) \rtimes (S_h \times S_g)$ (RoPE: $\mathrm{GL}(d_k) \mapsto C_{\mathrm{RoPE}}$), and we prove standard top-$k$ MoE routing is invariant to all gauge transformations.

## 1 Introduction

Neural networks often contain many different parameter settings that realize the same function. Such symmetries form high-dimensional equivalence classes in parameter space, undermining naive assumptions of unique optima and confounding measures like sharpness. In Transformers, this issue is particularly acute. Recent work has shown that standard metrics break down because Transformers possess rich symmetries that induce flat directions along which the network and its loss remain identical (da Silva et al., 2025; Zhang et al., 2025). These intrinsic symmetries arise from the architecture itself, not from training dynamics or regularization. Understanding them completely is crucial for developing better optimization procedures and a comprehensive theory for Transformers.

This paper establishes the complete parameter symmetry structure of Transformer architectures. We prove that for the canonical Transformer family—encompassing GPT-2 (Radford et al., 2019), BERT (Devlin et al., 2018), and their descendants—the gauge group of parameter transformations that preserve the function equals exactly $G_{\max} = ((\mathrm{GL}(d_k))^h \times (\mathrm{GL}(d_v))^h) \rtimes S_h$ on generic parameters. Here $h$ denotes the number of atten-

tion heads and $d_k, d_v$ represent query-key and value dimensions respectively. No additional symmetries exist beyond those in this group, establishing this characterization as complete and maximal.

The attention mechanism operates through two independent computational pipelines that each contain internal degrees of freedom. The attention scores depend only on the bilinear form $QK^\top = XW_Q(W_K)^\top X^\top$. Any transformation that preserves this product leaves the scores unchanged. Similarly, the value transformation depends only on the composed mapping $VW_O = XW_VW_O$. These observations lead directly to gauge symmetries: transforming $(W_Q, W_K) \mapsto (W_QA, W_K(A^{-1})^\top)$ preserves the query-key product, while $(W_V, W_O) \mapsto (W_VC, C^{-1}W_O)$ preserves the value-output composition, for any invertible matrices $A$ and $C$ of appropriate dimensions.

**Our Contributions.** We advance the understanding of Transformer geometry through four fundamental contributions: (1) **Complete gauge group characterization with proof of maximality.** We prove $G_{\max} = ((\mathrm{GL}(d_k))^h \times (\mathrm{GL}(d_v))^h) \rtimes S_h$ and establish that no additional symmetries exist through three independent arguments: Lie algebra characterization, attention weight identifiability, and necessary factorization of transformations. (2) **Extension to position-encoded architectures.** For RoPE and relative encodings, we show the query-key sector reduces to the commutant of position rotations (typically $(\mathrm{GL}(1, \mathbb{C}))^{d_k/2}$), yielding $G_{\mathrm{RoPE}} = ((\mathcal{C}_{\mathrm{RoPE}})^h \times (\mathrm{GL}(d_v))^h) \rtimes S_h$. (3) **Multi-layer direct product structure.** We prove residual connections and LayerNorm prevent inter-layer gauge coupling, yielding $G_{\mathrm{Model}} = \prod_{l=1}^{L} G_{\mathrm{Layer}}^{(l)}$. (4) **Practical implications for optimization and compression.** The gauge structure explains Hessian nullspaces with $h(d_k^2 + d_v^2)$ zero eigenvalues per layer, enables lossless compression eliminating millions of parameters, and provides the foundation for gauge-aware optimization algorithms. (5) **Architectural variants.** Beyond canonical MHA, we characterize the exact gauge group when query heads share K/V projections (GQA/MQA): the permissible Q/K and V transformations are tied per K/V group. For RoPE models, the Q/K sector reduces to the RoPE commutant per group. We also show that standard top-$k$ MoE routing preserves gauge equivalence, as the hidden states are unchanged by gauge transformations.

## 2 Transformer Architecture and Mathematical Framework

We establish notation and architectural conventions for the canonical Transformer family. We define the generic conditions under which global maximality holds and introduce the mathematical framework for analyzing both standard multi-head attention and position-encoded variants.

### 2.1 Multi-Head Attention Mechanism

The canonical Transformer family consists of architectures employing multi-head attention with softmax normalization, layer normalization, and feed-forward networks with smooth activation functions. This family encompasses GPT-2, BERT, GPT-3, and position-encoded variants such as LLaMA and Qwen. We adopt the row-vector convention where data flows as row vectors through matrices acting on the right.

For input $X \in \mathbb{R}^{n \times d_{\mathrm{model}}}$ representing a sequence of $n$ token embeddings, each attention head $i \in \{1, \ldots, h\}$ computes queries, keys, and values through linear projections:

$$Q_i = XW_Q^{(i)} \in \mathbb{R}^{n \times d_k}, \quad K_i = XW_K^{(i)} \in \mathbb{R}^{n \times d_k}, \quad V_i = XW_V^{(i)} \in \mathbb{R}^{n \times d_v} \tag{1}$$

where $W_Q^{(i)}, W_K^{(i)} \in \mathbb{R}^{d_{\mathrm{model}} \times d_k}$ and $W_V^{(i)} \in \mathbb{R}^{d_{\mathrm{model}} \times d_v}$.

The scaled dot-product attention for head $i$ is

$$A_i(X) = \mathrm{softmax}\left(\frac{Q_iK_i^\top}{\sqrt{d_k}}\right)V_i, \qquad A_i(X) \in \mathbb{R}^{n \times d_v}, \tag{2}$$

| ID | Mathematical Statement | Architectural Examples |
|----|------------------------|------------------------|
| A1 | Standard multi-head architecture, no weight sharing | All production models |
| A2 | $d_{\text{model}} = h \cdot d_v$ (canonical dimensions) | BERT-Base, GPT-2: $768 = 12 \times 64$ |
| A3 | Biases absent or transform covariantly | Standard implementations |
| A4 | $W_Q^{(i)}, W_K^{(i)}, W_V^{(i)}$ have full column rank | Generic stratum $\Theta_0$ |
| A5 | LayerNorm in standard Pre-LN or Post-LN blocks; placement does not break MHA gauge invariance (see Corollary 4.1) | Common implementations |
| A6 | $\text{rank}([W_Q^{(i)} \mid W_K^{(i)}]) = 2d_k$, $d_{\text{model}} \geq 2d_k$ | GPT-2: $768 \geq 128 = 2 \times 64$ |

Table 1: Assumptions defining the generic stratum where global maximality holds. Full justifications appear in Appendix A.

where $Q_i = XW_Q^{(i)}$, $K_i = XW_K^{(i)}$, $V_i = XW_V^{(i)}$, and the softmax acts row-wise (over tokens). The multi-head attention output is

$$\text{MHA}(X) \;=\; \left[A_1(X) \;\|\; \cdots \;\|\; A_h(X)\right] W_O \;=\; \sum_{i=1}^{h} A_i(X)\, W_{O,i}, \tag{3}$$

with $W_O \in \mathbb{R}^{(hd_v) \times d_{\text{model}}}$ partitioned into blocks $W_{O,i} \in \mathbb{R}^{d_v \times d_{\text{model}}}$.

## 2.2 Assumptions for Global Maximality

**Head-sharing variant of A2.** When $h$ query heads are partitioned into $g$ K/V groups by a surjection $m : \{1, \ldots, h\} \to \{1, \ldots, g\}$, we relax A2 to:

(A2′)  No two *distinct* K/V groups share exactly the same $K$ subspace or the same $V$ subspace;

within a group $r$, sharing $\left(W_K^{(r)}, W_V^{(r)}\right)$ across its query heads is by design and excluded from distinctness constraints.

The generic stratum $\Theta_0$ consists of parameter configurations satisfying assumptions A1-A6 (Table 1). This forms a Zariski-open dense subset of the full parameter space $\Theta$, meaning the exceptional set where these conditions fail has Lebesgue measure zero. The exceptional set forms a determinantal variety defined by the vanishing of certain minors of the projection matrices.

The critical assumption A6 requires that the stacked matrix $[W_Q^{(i)} \mid W_K^{(i)}]$ has full column rank $2d_k$ and $d_{\text{model}} \geq 2d_k$. This enables the head-wise attention controllability property: for each head, we can construct inputs that activate it selectively while suppressing others. This property holds generically and is essential for proving attention weight identifiability.

## 2.3 Position-Encoded Architectures

For architectures with rotary position embeddings (RoPE), position-dependent rotations are applied after linear projections:

$$Q_i^{\text{RoPE}} = Q_i R_{\text{pos}}, \quad K_i^{\text{RoPE}} = K_i R_{\text{pos}} \tag{4}$$

where $R_{\text{pos}} \in \text{SO}(d_k)$ encodes positional information through block-diagonal rotation matrices.

These transformations constrain the gauge group. Any gauge transformation $A \in \text{GL}(d_k)$ must commute with all position rotations to preserve attention computation, restricting query-key gauge freedom to:

$$\mathcal{C}_{\text{RoPE}} = \{A \in \text{GL}(d_k) \mid [A, R_{\text{pos}}] = 0 \text{ for all positions}\} \tag{5}$$

For standard RoPE with $2 \times 2$ rotation blocks, the commutant consists of block-diagonal matrices where each block has form $aI_2 + bJ$ with $J$ the 90° rotation. This yields $\mathcal{C}_{\text{RoPE}} \cong (\text{GL}(1, \mathbb{C}))^{d_k/2}$ with real dimension $d_k$.

**Grouped heads.** Under head sharing, the Q/K sector reduces per group: replace $\mathrm{GL}(d_k)$ by a single $A_r \in C_{\mathrm{RoPE}}$ for each K/V group $r$, giving a tied commutant $(C_{\mathrm{RoPE}})^g$ in place of $(\mathrm{GL}(d_k))^h$.

# 3 GLOBAL MAXIMALITY OF THE GAUGE GROUP

We now establish the complete gauge group structure for the canonical Transformer family. Our proof strategy proceeds through three stages: constructing the gauge group candidate, proving these transformations preserve the function, and establishing no additional symmetries exist.

**Definition 3.1** (Standard Gauge Transformations). *The standard gauge group $G_{\max}$ consists of transformations parametrized by $(A_i, C_i) \in \mathrm{GL}(d_k) \times \mathrm{GL}(d_v)$ for each head $i \in \{1, \dots, h\}$ and permutations $\sigma \in S_h$, acting as:*

$$(W_Q^{(i)}, W_K^{(i)}) \mapsto (W_Q^{(\sigma(i))} A_{\sigma(i)}, W_K^{(\sigma(i))} (A_{\sigma(i)}^{-1})^\top) \tag{6}$$

$$(W_V^{(i)}, W_{O,i}) \mapsto (W_V^{(\sigma(i))} C_{\sigma(i)}, C_{\sigma(i)}^{-1} W_{O,\sigma(i)}) \tag{7}$$

*We index $(A_i, C_i)$ after applying $\sigma$, giving the semidirect product structure $G_{\max} = ((\mathrm{GL}(d_k))^h \times (\mathrm{GL}(d_v))^h) \rtimes S_h$.*

**Theorem 3.2** (Gauge group under head sharing). *Let the $h$ query heads be partitioned into $g$ K/V groups via a surjection $m : \{1, \dots, h\} \to \{1, \dots, g\}$, with each group $r$ sharing $(W_K^{(r)}, W_V^{(r)})$. On the generic stratum satisfying A1–A6 with (A2) replaced by (A2'), the full parameter symmetry group of the MHA block is*

$$G_{\mathrm{share}} = \Big((\mathrm{GL}(d_k))^g \times (\mathrm{GL}(d_v))^g\Big) \rtimes (S_h \times S_g),$$

*where the continuous part acts by*

$$W_Q^{(i)} \mapsto W_Q^{(i)} A_{m(i)}, \quad W_K^{(r)} \mapsto W_K^{(r)} A_r^{-\top},$$
$$W_V^{(r)} \mapsto W_V^{(r)} C_r, \qquad W_O^{(i)} \mapsto C_{m(i)}^{-1} W_O^{(i)},$$

*with $A_r \in \mathrm{GL}(d_k)$ and $C_r \in \mathrm{GL}(d_v)$ chosen independently for each $r \in \{1, \dots, g\}$. The discrete factor $(S_h \times S_g)$ acts by independent relabelings of query heads and K/V groups (and the induced relabeling of $m$). No additional parameter symmetries exist.*

*Proof sketch.* Forward containment is by direct verification: with $A_r \in \mathrm{GL}(d_k)$ and $C_r \in \mathrm{GL}(d_v)$, $W_Q^{(i)} \mapsto W_Q^{(i)} A_{m(i)}$, $W_K^{(r)} \mapsto W_K^{(r)} A_r^{-\top}$, $W_V^{(r)} \mapsto W_V^{(r)} C_r$, $W_O^{(i)} \mapsto C_{m(i)}^{-1} W_O^{(i)}$ leaves attention scores and head outputs unchanged. For reverse containment, the same identifiability and block-diagonality steps used in the canonical maximality argument apply under (A2'): the preserved bilinear forms force stabilization of each $K$-subspace within its K/V group (tying $A_r$ per group), and equality of head outputs forces $C_r$ to be tied per group as well. Any mixing of heads beyond $S_h$ or of groups beyond $S_g$ would change the score bilinear forms or the value spans, hence is excluded. *Full proof in Appendix B.7.* □

**Corollary 3.3** (RoPE variant under head sharing). *For RoPE architectures, replace $\mathrm{GL}(d_k)$ by the RoPE commutant $C_{\mathrm{RoPE}}$ in Theorem 3.2, yielding*

$$G_{\mathrm{share,RoPE}} = \Big((C_{\mathrm{RoPE}})^g \times (\mathrm{GL}(d_v))^g\Big) \rtimes (S_h \times S_g).$$

Full proof in Appendix D.1.

**Corollary 3.4** (Gauge dimension under sharing). *For a single layer with head sharing, the continuous component of $G_{\mathrm{share}}$ has real dimension*

$$\dim_{\mathbb{R}}\big((\mathrm{GL}(d_k))^g \times (\mathrm{GL}(d_v))^g\big) = g\,(d_k^2 + d_v^2),$$

*while the discrete factor $(S_h \times S_g)$ contributes zero. In RoPE models, replacing $\mathrm{GL}(d_k)$ by the commutant $C_{\mathrm{RoPE}}$ yields*

$$\dim_{\mathbb{R}}\big((C_{\mathrm{RoPE}})^g \times (\mathrm{GL}(d_v))^g\big) = g\,(\dim_{\mathbb{R}} C_{\mathrm{RoPE}} + d_v^2),$$

*and for the standard RoPE construction $C_{\mathrm{RoPE}} \cong (\mathrm{GL}(1, \mathbb{C}))^{d_k/2}$ so $\dim_{\mathbb{R}} C_{\mathrm{RoPE}} = d_k$, giving $g\,(d_k + d_v^2)$.* Full proof in Appendix D.2.

**Corollary 3.5** (MoE router invariance). *Let a standard top-k router compute logits $\rho = W_r h_t + b$ from the block hidden state $h_t$. Since gauge transformations preserve each head output and hence $h_t$, the router logits and expert selection are unchanged under all $g \in G_{\max}$, $G_{\text{share}}$, or $G_{\text{share,RoPE}}$.* Full proof in Appendix C.4.

**Theorem 3.6** (Global Maximality on the Generic Stratum). *For the canonical Transformer family satisfying assumptions A1-A6, the gauge group on the generic stratum $\Theta_0$ equals exactly $G_{\max} = ((\text{GL}(d_k))^h \times (\text{GL}(d_v))^h) \rtimes S_h$. No additional parameter symmetries exist beyond those in this group.* Complete proof in Appendices B.1–B.6.

**Corollary 3.7** (Gauge Dimension). *The continuous gauge group dimension for canonical Transformers is $h(d_k^2 + d_v^2)$ per layer. For standard configurations with $h = 12$ heads and $d_k = d_v = 64$, this yields $12 \times (64^2 + 64^2) = 98,304$ continuous degrees of freedom per attention layer.* Dimension count in Appendix B.7.

**Corollary 3.8** (Gauge Group for RoPE Architectures). *For architectures with rotary position embeddings, the gauge group becomes $G_{RoPE} = ((\mathcal{C}_{RoPE})^h \times (\text{GL}(d_v))^h) \rtimes S_h$ where $\mathcal{C}_{RoPE}$ is the commutant of position rotations. For standard RoPE with $2 \times 2$ rotation blocks, $\mathcal{C}_{RoPE} \cong (\text{GL}(1, \mathbb{C}))^{d_k/2}$ has real dimension $d_k$, reducing gauge freedom from $d_k^2$ to $d_k$ per head in the query-key sector.* See Appendix D.1 (commutant) and Appendix D.2 (group reduction).

# 4 Architectural Extensions and Multi-Layer Structure

Having established global maximality for multi-head attention, we extend our analysis to complete Transformer architectures. We prove that LayerNorm preserves gauge symmetries, characterize the combined attention-FFN block structure, and establish that multi-layer Transformers have a direct product gauge group with no inter-layer coupling.

## 4.1 LayerNorm Preserves Gauge Symmetry

**Corollary 4.1** (LayerNorm compatibility). *Since MHA's output is gauge-invariant and LayerNorm acts on that tensor (Post-LN: $X + \text{MHA}(X)$; Pre-LN: $\text{LN}(X)$), LayerNorm preserves the symmetry in both variants; see Appendix C.2.* Details in Appendix C.5.

## 4.2 Feed-Forward Networks and Block Structure

Feed-forward networks in Transformer blocks possess limited symmetry structure due to non-linear activations.

**Lemma 4.2** (FFN Gauge Structure). *The gauge group of the feed-forward network with GELU activation consists only of hidden unit permutations: $G_{FFN} = S_{d_{ff}}$ where $d_{ff}$ is the hidden dimension.* Proof in Appendix C.2.

Since MHA and FFN operate on disjoint parameters and residual connections prevent coupling, their gauge groups combine as a direct product: $G_{\text{Block}} = G_{\text{MHA}} \times G_{\text{FFN}}$.

## 4.3 Multi-Layer Direct Product Structure

**Theorem 4.3** (Direct Product for Multi-Layer Transformers). *For an $L$-layer Transformer, the gauge group factors as*

$$G_{\text{Model}} = \prod_{l=1}^{L} G_{\text{Block}}^{(l)}, \tag{8}$$

*where each $G_{\text{Block}}^{(l)} = G_{\text{MHA}}^{(l)} \times G_{\text{FFN}}^{(l)}$ acts only on the parameters of layer $l$.* Proof in Appendix C.3.

**Proof sketch.** Inter-layer coupling is ruled out by two facts. First, the residual connection forces any transformation on the block output at layer $l$ to apply identically to

the untouched residual stream; this is impossible unless the transformation is the identity. Second, LayerNorm is not equivariant to general linear scalings: for $M \neq I$,

$$\mathrm{LN}(Mx) \;\neq\; M\,\mathrm{LN}(x),$$

since $M$ alters feature statistics non-uniformly. Consequently, a layer's gauge action cannot propagate across layers, yielding the layerwise direct product.

**Remark 4.4** (Multi-layer structure with head sharing). *All statements in §4 remain valid when $G_{\mathrm{MHA}}$ is replaced by $G_{\mathrm{share}}$ (or $G_{\mathrm{share,RoPE}}$ in RoPE models): the per-layer gauge group factors independently and no inter-layer coupling appears.*

## 5 Implications for Optimization and Loss Landscape Geometry

The gauge structure has direct consequences for optimization geometry, creating flat directions in the loss landscape and explaining empirical phenomena in Transformer training.

### 5.1 Hessian Nullspace Structure

**Proposition 5.1** (Hessian Nullspace from Gauge Symmetry). *At any critical point $\theta^* \in \Theta_0$, the Hessian $\nabla^2 L(\theta^*)$ has nullspace dimension at least $h(d_k^2 + d_v^2)$ per layer, with null directions corresponding to the Lie algebra $\mathfrak{g}_{\max}$ of gauge transformations.* Proof in Appendix E.6.

Since gauge transformations preserve the network function, they preserve the loss: $L(g(\theta)) = L(\theta)$ for all $g \in G_{\max}$. Along any one-parameter subgroup $g_t = \exp(tX)$ where $X \in \mathfrak{g}_{\max}$, the loss remains constant, yielding zero curvature in these directions. For an $L$-layer model, the nullspace dimension is at least $L \cdot h(d_k^2 + d_v^2)$.

**Corollary 5.2** (Hessian nullspace under head sharing). *At any critical point on $\Theta_0$ with head sharing, $\nabla^2 L(\theta^*)$ has nullspace dimension at least $g(d_k^2 + d_v^2)$ per layer in the canonical case and $g(\dim_{\mathbb{R}} C_{\mathrm{RoPE}} + d_v^2)$ for RoPE. For an $L$-layer model, the lower bound scales as $L$ times these quantities.* Full proof in Appendix E.5.

### 5.2 Optimization in Quotient Space

The gradient $\nabla L(\theta)$ is orthogonal to gauge orbits at every point, so gradient descent naturally respects the gauge without explicit constraints. Optimization effectively proceeds on the quotient $\Theta_0/G_{\max}$ of gauge-inequivalent configurations; its effective dimension is $\dim_{\mathrm{eff}} = \dim(\Theta) - L\,h(d_k^2 + d_v^2)$.

This dimension reduction partially explains why Transformers with hundreds of millions of parameters can be trained effectively—the true degrees of freedom are substantially fewer than the raw parameter count.

### 5.3 Mode Connectivity and Flat Minima

Critical points of Transformer training form continuous manifolds rather than isolated points. Within each connected component of the gauge orbit, all points represent identical functions with identical loss. This provides a geometric explanation for the prevalence of flat minima observed empirically and the phenomenon of linear mode connectivity between independently trained models. Two parameter configurations that differ only by a gauge transformation are connected by a flat path in the loss landscape with constant loss value.

The gauge structure also explains why standard sharpness measures based on Hessian eigenvalues are problematic for Transformers. The Hessian necessarily has at least $L \cdot h(d_k^2 + d_v^2)$ zero eigenvalues from gauge directions. Meaningful sharpness must be measured on the quotient space or restricted to gauge-orthogonal directions to avoid including these artificial flat directions.

## 6  EXPERIMENTAL VALIDATION

We validate our theoretical predictions through systematic experiments confirming that gauge transformations preserve network outputs to numerical precision while transformations outside $G_{\max}$ produce substantial changes. Our experiments test diverse architectures including non-square dimensions $(d_k, d_v) \in \{(64, 64), (64, 128), (128, 64)\}$ to verify sector independence, trained GPT-2 checkpoints to confirm post-optimization invariance, and explicit Hessian nullspace validation confirming $h(d_k^2 + d_v^2)$ zero eigenvalues. Complete protocols and extended results appear in Appendix F.

We test $10^5$ trials consisting of 1,000 random inputs with 100 random gauge transformations each, using IEEE double-precision arithmetic (float64). The relative Frobenius error $\|Y' - Y\|_F / (\|Y\|_F + \varepsilon_{\mathrm{mach}})$ quantifies differences between original and transformed outputs, where $\varepsilon_{\mathrm{mach}} \approx 2.22 \times 10^{-16}$ is machine epsilon. All experiments use fixed random seeds, disabled dropout, and LayerNorm in evaluation mode to ensure deterministic and reproducible results.

For transformations within $G_{\max}$, we observe mean errors of 2-4 $\times 10^{-15}$ and maximum error of $5.28 \times 10^{-15}$, corresponding to approximately 9-24$\varepsilon_{\mathrm{mach}}$. This level of preservation confirms exact functional invariance within finite-precision arithmetic limits. The errors fall within the expected accumulation band of 10-1000$\varepsilon_{\mathrm{mach}}$ for complex matrix operations involving multiple multiplications and additions.

Query-key and value-output transformations, tested separately and in composition, produce errors within the same 10-25$\varepsilon_{\mathrm{mach}}$ band, confirming the direct product structure. Specifically, query-key transformations alone yield maximum errors of $5.61 \times 10^{-16}$ (approximately 2.5$\varepsilon_{\mathrm{mach}}$), while value-output transformations produce maximum errors of $4.61 \times 10^{-15}$ (approximately 21$\varepsilon_{\mathrm{mach}}$). When both are composed, the errors remain bounded, demonstrating no accumulation from sector interaction.

Invalid transformations such as non-block-diagonal mixing across heads produce $O(1)$ relative changes with median approximately 1.0 and 95th percentile exceeding 1.1, providing decisive evidence for maximality. The dramatic difference between valid transformations (errors at 10-25$\varepsilon_{\mathrm{mach}}$) and invalid ones (errors at $O(1)$) spans over 14 orders of magnitude, empirically supporting that no additional symmetries exist beyond those in $G_{\max}$.

Results remain consistent across architectures with $h \in \{4, 8, 12\}$ heads and dimensions $d_k = d_v \in \{64, 128\}$. Computational time scales linearly with model complexity, requiring approximately 170 seconds for the most complex configuration on an NVIDIA H100 GPU. Complete methodology, statistical analysis, and hardware specifications appear in Appendix F.

## 7  PRACTICAL IMPLICATIONS AND APPLICATIONS

The gauge structure enables several practical applications in model compression, optimization, and analysis.

### 7.1  QUANTIFYING PARAMETER REDUNDANCY

A standard Transformer Base model with 110M parameters contains approximately 1.18M continuously redundant dimensions from gauge symmetries, representing 1.1% of total parameters. While modest as a percentage, this corresponds to over one million parameters that can be varied without changing the function, fundamentally affecting optimization geometry. For larger models, the redundancy scales proportionally. GPT-3 scale models with 175B parameters contain over 300M redundant dimensions. The redundancy ratio $h(d_k^2 + d_v^2)/d_{\mathrm{total}}$ decreases with model scale when head dimensions remain fixed while the number of heads increases, as is common in production architectures. This scaling property suggests that architectural innovations reducing gauge redundancy while preserving expressiveness could lead to more parameter-efficient models.

## 7.2 Model Compression via Gauge-Fixing

Gauge-fixing to canonical form enables lossless compression by eliminating $h(d_k^2 + d_v^2)$ parameters per layer without any approximation. The canonical form can be chosen to balance query-key Gram matrices $(W_Q^{(i)})^\top W_Q^{(i)} = (W_K^{(i)})^\top W_K^{(i)}$, orthonormalize value projections $(W_V^{(i)})^\top W_V^{(i)} = I_{d_v}$, and order heads by norm to break permutation symmetry.

This compression is exact, preserving the network function perfectly while reducing parameter count. The computational cost of gauge-fixing is $\mathcal{O}(h(d_k^3 + d_v^3))$ per layer, negligible compared to inference cost. Implementation algorithms and complexity analysis appear in Appendix G.1.

## 7.3 Gauge-Aware Optimization

Standard optimizers waste computation updating along gauge directions where the function remains unchanged. Projecting gradients onto gauge-orthogonal subspaces eliminates these redundant updates, potentially accelerating convergence. The projection can be implemented efficiently using the structure of $\mathfrak{g}_{\max}$:

$$\nabla_{\mathrm{proj}} L = \nabla L - \Pi_{\mathfrak{g}} \nabla L \qquad (9)$$

where $\Pi_{\mathfrak{g}}$ denotes projection onto the gauge tangent space. This adds negligible computational overhead while focusing optimization on function-changing directions.

## 7.4 Model Merging and Averaging

Independent training yields different gauge-equivalent representations even when models converge to similar functions. Naive parameter averaging fails because models occupy different points in the same gauge orbit. Gauge alignment before averaging enables meaningful parameter interpolation by transforming models to a common gauge. The alignment procedure involves solving Procrustes problems for optimal $(A_i, C_i)$ per head and finding the optimal head permutation via the Hungarian algorithm. This explains why sophisticated model merging techniques succeed where simple averaging fails, and provides a principled framework for developing improved merging algorithms.

## 7.5 Implications for Architecture Design

The gauge structure provides insights for architectural innovations. Multi-query attention and grouped-query attention deliberately couple heads by sharing key-value projections, reducing gauge freedom at the cost of expressiveness. This trade-off can now be quantified precisely through the dimension of the resulting gauge group. Future architectures might exploit gauge structure more deliberately, perhaps by operating directly on gauge-invariant features or incorporating gauge-aware regularization during training.

**Gauge dimension with sharing.** If $h$ queries share $g$ K/V groups, the continuous gauge dimension per layer reduces from $h(d_k^2 + d_v^2)$ to $g(d_k^2 + d_v^2)$ (RoPE: $g(d_k + d_v^2)$), precisely quantifying the expressiveness–symmetry trade-off induced by head sharing.

## 8 Related Work

Parameter symmetries in neural networks have been studied extensively. For fully connected networks, permutation symmetries arising from neuron reordering create $S_n$ symmetry groups (Hecht-Nielsen, 1990; Kurková & Kainen, 1994). These discrete symmetries are substantially smaller than the continuous gauge groups we identify in Transformers. Convolutional networks exhibit translation invariance and filter permutation symmetries (LeCun et al., 1998; Cohen & Welling, 2016).

For Transformers specifically, recent work has begun identifying partial symmetries. van Nierop (2024) identified gauge invariance from a physics perspective without proving completeness. Entezari et al. (2022) studied permutation invariance for mode connectivity.

da Silva et al. (2025); Zhang et al. (2025) demonstrated symmetry effects on sharpness and model fusion. Henry et al. (2025) analyzed simplified attention without softmax, which exhibits fundamentally different geometry.

Previous optimization work recognized symmetry-induced flat directions (Kunin et al., 2020) and mode connectivity (Garipov et al., 2018; Draxler et al., 2018). Model merging techniques (Ainsworth et al., 2023; Wortsman et al., 2022) implicitly leverage symmetries through weight matching. Our complete characterization with maximality proof provides the theoretical foundation for these observations, explaining why certain techniques succeed and suggesting systematic improvements.

## 9 Discussion and Future Directions

Our characterization opens several research directions. Gauge-aware optimization algorithms that explicitly project out redundant directions could accelerate training. Theoretical analysis of how gauge structure affects generalization may yield new complexity measures beyond parameter counting. Understanding how architectural innovations like mixture-of-experts or state-space models modify gauge structure could guide design choices. The reduction in gauge freedom from RoPE architectures suggests a connection between symmetry constraints and parameter efficiency. Architectures that deliberately break certain symmetries while preserving others might achieve better trade-offs between expressiveness and efficiency. The gauge perspective also suggests new initialization schemes that distribute parameters uniformly across gauge orbits rather than in raw parameter space.

**Beyond MHA.** Modules that consume the block hidden state (e.g., top-$k$ MoE routers) inherit gauge invariance because $h_t$ is preserved; extending the taxonomy of such "downstream-invariant" modules is a promising direction.

**Practical implications.** Our classification of maximal gauge symmetries is upstream of both serving and verification. First, training-free gauge canonicalization (orthonormal values; balanced queries/keys; RoPE-commutant tying) yields KV caches in a compression-friendly basis, composes multiplicatively with head sharing in GQA/MQA (factor $h/g$), and preserves model function exactly. Second, for verifiable inference, any prover/constraint cost that scales with K/V heads inherits the same layerwise factorization and the $(h/g)$ structural reduction; MoE routing remains invariant because hidden states are unchanged by gauge actions. Finally, LayerNorm compatibility and the depthwise direct-product group ensure safe, layer-local deployment (e.g., per-layer rewrites or rolling upgrades). Concurrent systems work explores these instantiations in serving and verification settings; our focus here is the symmetry structure that enables them.

## 10 Conclusion

We have established the complete gauge group structure of Transformer architectures, proving global maximality for the canonical family under mild generic conditions. The gauge group $G_{\max} = ((\mathrm{GL}(d_k))^h \times (\mathrm{GL}(d_v))^h) \rtimes S_h$ captures all parameter redundancies in standard multi-head attention, with no additional symmetries beyond those we identify. This definitive characterization resolves the fundamental question of parameter-function correspondence for Transformers.

Our proof of maximality through three independent mathematical arguments—Lie algebra characterization, attention weight identifiability, and necessary factorization—ensures completeness. These results extend through LayerNorm and residual connections, yielding the direct product structure $G_{\text{Model}} = \prod_{l=1}^{L} G_{\text{Block}}^{(l)}$ for multi-layer architectures.

The gauge structure reveals substantial parameter redundancy exceeding one million continuous degrees of freedom in standard models. These redundant directions correspond to flat regions in the loss landscape, explaining empirical phenomena and enabling practical advances in compression, optimization, and model merging. As Transformers continue driving advances in artificial intelligence, leveraging their mathematical structure becomes increasingly critical for both theoretical understanding and practical improvements.

**Ethics Statement.** We have read and will adhere to the ICLR Code of Ethics. This work is a theoretical and post-hoc reparameterization study of existing Transformer architectures; it does not involve human subjects, sensitive attributes, or personally identifiable information. All models and datasets referenced are publicly available and used under their respective licenses (listed in the appendix). As with any efficiency or analysis tool, model transformations could be misapplied; we recommend standard safety practices (content filters, rate limits, and monitoring) when deploying any derived systems.

**Reproducibility Statement.** We provide complete details to reproduce our experiments: architectures and checkpoints, dataset sources and preprocessing, exact hyperparameters, determinism settings (fixed seeds; controlled precision), and hardware/software versions. We include scripts to (i) perform layerwise gauge transformations and canonicalization (QR on $W_V$, geometric-mean balancing of $Q/K$, RoPE-commutant projection, deterministic head permutation), (ii) verify functional invariance to numerical precision, and (iii) regenerate all tables and figures from a fresh checkout.

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

# A  Mathematical Framework and Complete Assumptions

This appendix provides complete justifications for all assumptions and establishes the mathematical framework underlying our gauge group characterization.

## A.1  Justification of Assumption A6: Head-wise Attention Controllability

**Proposition A.1** (Attention Controllability on Generic Parameters). *Under assumptions A1-A5, the head-wise attention controllability condition A6 holds on a Zariski-open dense subset of the parameter space where projection matrices have full column rank.*

*Proof.* Consider the attention weight map $\Phi_i : \mathbb{R}^{n \times d_{\text{model}}} \to \mathbb{R}^{n \times n}$ defined by $\Phi_i(X) = \text{softmax}(XW_Q^{(i)}(XW_K^{(i)})^\top / \sqrt{d_k})$. For generic parameters where $W_Q^{(i)}$ and $W_K^{(i)}$ have full column rank and their column spaces are in general position relative to other heads, we construct the required inputs as follows.

For head $i$, choose $X^{(i)} = [v_1 \cdots v_n]^\top$ where $v_j \in \mathbb{R}^{d_{\text{model}}}$ are constructed such that $v_j W_Q^{(i)} = e_j \sqrt{d_k}$ and $v_j W_K^{(i)} = e_j \sqrt{d_k}$, where $e_j$ is the $j$-th standard basis vector in $\mathbb{R}^{d_k}$. This linear system is generically solvable provided the stacked matrix $[W_Q^{(i)} \ W_K^{(i)}]$ has full column rank $2d_k$ (a determinantal/Zariski-open condition) and $d_{\text{model}} \geq 2d_k$. This dimensional requirement is the exact condition needed for the head-isolation construction. For reference, canonical GPT-2 satisfies this with margin: $d_{\text{model}} = 768 \geq 128 = 2 \times 64 = 2d_k$. We take $n \leq d_k$. This construction yields $(XW_Q^{(i)})(XW_K^{(i)})^\top = d_k I_n$. For any $\lambda > 0$, replace $X^{(i)}$ by $\lambda X^{(i)}$. Then $Q_i K_i^\top / \sqrt{d_k} = \lambda^2 I_n$, and $\alpha_i(\lambda X^{(i)}) = \text{softmax}(\lambda^2 I_n) \to I_n$ as $\lambda \to \infty$.

For heads $j \neq i$, since the column spaces of $(W_Q^{(j)}, W_K^{(j)})$ and $(W_Q^{(i)}, W_K^{(i)})$ are in general position for generic parameters, we can choose the vectors $v_j$ to lie predominantly in the orthogonal complement of the column space of $[W_Q^{(j)} | W_K^{(j)}]^\top$. This yields $\|XW_Q^{(j)}\|_F, \|XW_K^{(j)}\|_F = O(\delta)$ for arbitrarily small $\delta > 0$, resulting in near-zero attention weights after softmax normalization.

The set where this construction fails consists of parameter configurations where the column spaces of different heads' projection matrices have non-trivial intersection. This constraint defines an algebraic variety of codimension at least 1, hence has Lebesgue measure zero in the parameter space. Therefore, A6 holds generically on the A4 stratum (i.e., on a Zariski-open dense subset of the full parameter space where the projections have full column rank). $\square$

**Proposition A.2** (Covariant Bias Transformation). *Requiring invariance of $(XW_Q^{(i)} + b_Q^{(i)})(XW_K^{(i)} + b_K^{(i)})^\top$ under gauge transformations forces $b_Q^{(i)} \mapsto b_Q^{(i)} A_i$ and $b_K^{(i)} \mapsto b_K^{(i)} (A_i^{-1})^\top$. Any other transformation creates uncancellable cross terms.*

*Proof.* Under the gauge transformation $(W_Q^{(i)}, W_K^{(i)}) \mapsto (W_Q^{(i)} A_i, W_K^{(i)} (A_i^{-1})^\top)$, the bilinear form with biases becomes:

$$(XW_Q^{(i)} A_i + b_Q'^{(i)})(XW_K^{(i)} (A_i^{-1})^\top + b_K'^{(i)})^\top \tag{10}$$

$$= XW_Q^{(i)} A_i (A_i^{-1}) W_K^{(i)\top} X^\top + XW_Q^{(i)} A_i (b_K'^{(i)})^\top + b_Q'^{(i)} (A_i^{-1}) W_K^{(i)\top})^\top X^\top + b_Q'^{(i)} (b_K'^{(i)})^\top \tag{11}$$

For this to equal the original $(XW_Q^{(i)} + b_Q^{(i)})(XW_K^{(i)} + b_K^{(i)})^\top$, we need:

$$XW_Q^{(i)} A_i (b_K'^{(i)})^\top = XW_Q^{(i)} (b_K^{(i)})^\top \tag{12}$$

$$b_Q'^{(i)} (A_i^{-1}) W_K^{(i)\top})^\top X^\top = b_Q^{(i)} (W_K^{(i)})^\top X^\top \tag{13}$$

$$b_Q'^{(i)} (b_K'^{(i)})^\top = b_Q^{(i)} (b_K^{(i)})^\top \tag{14}$$

Since these must hold for all $X$, we require:

$$A_i (b_K'^{(i)})^\top = (b_K^{(i)})^\top \implies b_K'^{(i)} = b_K^{(i)} (A_i^{-1})^\top \tag{15}$$

$$b_Q'^{(i)} (A_i^{-1}) = b_Q^{(i)} \implies b_Q'^{(i)} = b_Q^{(i)} A_i \tag{16}$$

These transformations automatically satisfy the third constraint, confirming the unique covariant transformation law. $\square$

**Remark A.3** (Measure-Theoretic Precision). *Throughout this paper, when we claim a property holds "generically" or that an exceptional set has "measure zero," we refer to the Lebesgue measure on the parameter space $\Theta \subset \mathbb{R}^D$ where $D$ denotes the total parameter dimension. The generic stratum $\Theta_0$ has full Lebesgue measure in $\Theta$, meaning $\mu(\Theta \setminus \Theta_0) = 0$ where $\mu$ denotes the Lebesgue measure. The exceptional set forms a determinantal variety defined by the vanishing of certain minors of the projection matrices.*

## A.2 Extended Geometric Intuition

**Example A.4** (Concrete Gauge Transformation with Complete Calculations). *For $W_Q = I_2$ and $W_K = \begin{bmatrix} 2 & 1 \\ 1 & 3 \end{bmatrix}$, consider the transformation with $A = \begin{bmatrix} 2 & 1 \\ 0 & 1 \end{bmatrix}$. Setting $W_Q' = W_Q A$ and $W_K' = W_K (A^{-1})^\top$ yields:*

*First, compute $A^{-1} = \begin{bmatrix} 1/2 & -1/2 \\ 0 & 1 \end{bmatrix}$ and $(A^{-1})^\top = \begin{bmatrix} 1/2 & 0 \\ -1/2 & 1 \end{bmatrix}$.*

*Then:*

$$W_Q' = I_2 \cdot \begin{bmatrix} 2 & 1 \\ 0 & 1 \end{bmatrix} = \begin{bmatrix} 2 & 1 \\ 0 & 1 \end{bmatrix} \tag{17}$$

$$W_K' = \begin{bmatrix} 2 & 1 \\ 1 & 3 \end{bmatrix} \cdot \begin{bmatrix} 1/2 & 0 \\ -1/2 & 1 \end{bmatrix} = \begin{bmatrix} 1/2 & 1 \\ -1 & 3 \end{bmatrix} \tag{18}$$

*Direct computation verifies:*

$$W_Q' (W_K')^\top = \begin{bmatrix} 2 & 1 \\ 0 & 1 \end{bmatrix} \cdot \begin{bmatrix} 1/2 & -1 \\ 1 & 3 \end{bmatrix} = \begin{bmatrix} 2 & 1 \\ 1 & 3 \end{bmatrix} = W_Q W_K^\top \tag{19}$$

*The attention scores remain identical despite the parameter transformation.*

## A.3 Technical Remarks and Contextual Notes

**Remark A.5** (Architectural Scope). *Global maximality requires $d_{model} = h \cdot d_v$, satisfied by BERT-Base (*$768 = 12 \times 64$*), GPT-2 (*$768 = 12 \times 64$*), and GPT-3 (*$12,288 = 96 \times 128$*). For architectures where this dimensional relationship does not hold exactly, continuous maximality of the gauge group is established, though the discrete structure may differ from the pure permutation group $S_h$.*

**Remark A.6** (Generic Stratum and Optimization Trajectories). *The generic stratum $\Theta_0$ where our results hold has full Lebesgue measure in the parameter space. Since optimization trajectories are absolutely continuous with respect to Lebesgue measure, they avoid the measure-zero exceptional set with probability one. This means our characterization applies to essentially all parameter configurations encountered during training, not just at initialization.*

**Remark A.7** (Essential Nature of Architectural Constraints). *Several architectural choices that appear conventional are actually mathematically essential for gauge symmetry preservation:*

1. *LayerNorm placement after the output projection $W_O$ rather than before is necessary. If LayerNorm operated on concatenated head outputs before projection, the value-output transformations $C_i$ would change individual head statistics, breaking gauge invariance.*

2. *The absence of weight sharing across heads is crucial. Shared parameters would couple gauge transformations across heads, preventing the direct product structure.*

3. *The standard scaling factor $1/\sqrt{d_k}$ in attention, while motivated by gradient stability, also ensures the gauge transformations preserve the softmax distribution uniformly.*

**Remark A.8** (Practical Implications of Gauge Structure). *The extensive flat directions from gauge symmetry help explain several empirical phenomena:*

- *The prevalence of wide minima in Transformer optimization despite high parameter counts*

- *Success of model averaging after gauge alignment (Izmailov et al., 2018; Wortsman et al., 2022)*

- *Effectiveness of lottery ticket pruning within gauge equivalence classes*

- *Convergence to similar performance from diverse initializations*

*These observations suggest that apparent complexity in parameter space masks geometric simplicity in function space.*

## A.4 Head Sharing: Notation and Genericity of (A2′)

We partition the $h$ query heads into $g$ K/V groups via a surjection $m : \{1, \ldots, h\} \to \{1, \ldots, g\}$; group $r$ shares parameters $(W_K^{(r)}, W_V^{(r)})$. For a head $i$, write $r = m(i)$.

**Assumption (A2′).** No two *distinct* K/V groups share exactly the same $K$ subspace or the same $V$ subspace; within a group $r$, sharing $(W_K^{(r)}, W_V^{(r)})$ across its query heads is by design and excluded from distinctness constraints.

**Proposition A.9** (Genericity of (A2′)). *Under any absolutely continuous weight initialization (w.r.t. Lebesgue measure), the set of parameters violating (A2′) has measure zero. In particular, with probability one, for all $r \neq s$, $\mathrm{col}(W_K^{(r)}) \neq \mathrm{col}(W_K^{(s)})$ and $\mathrm{col}(W_V^{(r)}) \neq \mathrm{col}(W_V^{(s)})$.*

*Proof.* Fix $r \neq s$. The event $\mathrm{col}(W_K^{(r)}) = \mathrm{col}(W_K^{(s)})$ imposes polynomial equalities among entries of $W_K^{(r)}, W_K^{(s)}$ (equality of Plücker coordinates), which defines a strict algebraic sub-

variety and thus a measure-zero set. A finite union over pairs $(r, s)$ remains measure zero. The argument for $V$ is identical. □

# B  COMPLETE PROOFS FOR GAUGE GROUP MAXIMALITY

## B.1  PROOF OF SUFFICIENCY

**Lemma B.1** (Every transformation in $G_{\max}$ preserves MHA). *For all $g \in G_{\max}$, $\theta \in \Theta_0$, and inputs $X \in \mathbb{R}^{n \times d_{model}}$, we have $MHA(X; g(\theta)) = MHA(X; \theta)$.*

*Proof.* Consider transformations with identity permutation first. For each head $i$, the query-key transformation preserves attention scores:

$$Q_i'(K_i')^\top = XW_Q^{(i)}A_i \cdot (A_i^{-1})^\top (W_K^{(i)})^\top X^\top \tag{20}$$

$$= XW_Q^{(i)}A_i(A_i^{-1})^\top (W_K^{(i)})^\top X^\top \tag{21}$$

$$= XW_Q^{(i)}I_{d_k}(W_K^{(i)})^\top X^\top \tag{22}$$

$$= XW_Q^{(i)}(W_K^{(i)})^\top X^\top = Q_iK_i^\top \tag{23}$$

Since softmax operates element-wise on each row, the attention weights $\alpha_i(X) = \text{softmax}(Q_iK_i^\top / \sqrt{d_k})$ remain unchanged.

The value-output transformation preserves each head's contribution:

$$A_i'(X)W_{O,i}' = \alpha_i(X)V_i'W_{O,i}' \tag{24}$$

$$= \alpha_i(X)XW_V^{(i)}C_i \cdot C_i^{-1}W_{O,i} \tag{25}$$

$$= \alpha_i(X)XW_V^{(i)}W_{O,i} = A_i(X)W_{O,i} \tag{26}$$

For permutations $\sigma \in S_h$, we have:

$$\text{MHA}(X; g(\theta)) = \sum_{i=1}^{h} A_{\sigma^{-1}(i)}(X)W_{O,\sigma^{-1}(i)} \tag{27}$$

$$= \sum_{j=1}^{h} A_j(X)W_{O,j} = \text{MHA}(X; \theta) \tag{28}$$

where we substituted $j = \sigma^{-1}(i)$. □

## B.2  PROOF OF ATTENTION WEIGHT IDENTIFIABILITY

**Lemma B.2** (Attention weights preserved up to permutation). *If $MHA(X; \theta') = MHA(X; \theta)$ for all inputs $X$, then there exists a permutation $\sigma \in S_h$ such that $\alpha_i(X; \theta') = \alpha_{\sigma(i)}(X; \theta)$ for all $X$ and all heads $i$.*

*Proof.* Using the head isolation property from assumption A6, for each head $i$ and any $\varepsilon > 0$, we can construct inputs $X^{(i)}$ such that:

$$\|\alpha_i(X^{(i)}) - I_n\|_F < \varepsilon, \quad \|\alpha_j(X^{(i)})\|_F < \varepsilon \text{ for all } j \neq i \tag{29}$$

For such inputs, the MHA output reduces to:

$$\text{MHA}(X^{(i)}) = \sum_{j=1}^{h} \alpha_j(X^{(i)})V_jW_{O,j} \approx \alpha_i(X^{(i)})V_iW_{O,i} + O(\varepsilon) \tag{30}$$

As $\varepsilon \to 0$, we have $\text{MHA}(X^{(i)}) \to V_iW_{O,i}$ where $V_i = X^{(i)}W_V^{(i)}$.

If the transformed parameters $\theta'$ yielded a different attention pattern structure not related by permutation, then for some head $i$, either no head in $\theta'$ approximates the identity on $X^{(i)}$, multiple heads in $\theta'$ approximate the identity on $X^{(i)}$, or a different head $j \neq i$ approximates the identity on $X^{(i)}$.

In the first two cases, $\mathrm{MHA}(X^{(i)}; \theta')$ would be a non-trivial mixture of multiple heads' values, contradicting $\mathrm{MHA}(X^{(i)}; \theta') = \mathrm{MHA}(X^{(i)}; \theta) \approx V_i W_{O,i}$.

The third case implies a bijection between heads, which must be a permutation $\sigma \in S_h$.

By continuity of the attention mechanism and density of isolating inputs (established in Proposition A.1), this permutation correspondence extends to all inputs $X$. $\qquad\square$

### B.3   Proof of Lie Algebra Characterization

**Lemma B.3** (Lie algebra equals $\mathfrak{g}_{\max}$). *For $\theta \in \Theta_0$, the Lie algebra of the gauge group $G(\theta)$ equals $\mathfrak{g}_{\max} = \bigoplus_{i=1}^{h} \mathfrak{gl}(d_k) \oplus \bigoplus_{i=1}^{h} \mathfrak{gl}(d_v)$.*

*Proof.* By Lemma B.2, attention weights are preserved up to a fixed permutation under any gauge transformation. Consider a smooth one-parameter family $g_t : \Theta_0 \to \Theta_0$ of gauge transformations with $g_0 = \mathrm{id}$. The invariance condition $\mathrm{MHA}(X; g_t(\theta)) = \mathrm{MHA}(X; \theta)$ holds for all $X$ and $t$.

Since attention weights are preserved (up to the fixed permutation), the first-order invariance splits into separate conditions for each head: $\delta(Q_i K_i^\top) = 0$ and $\delta(V_i W_{O,i}) = 0$.

For the query-key sector, differentiating at $t = 0$:

$$\frac{d}{dt}\Big|_{t=0} Q_i(t) K_i(t)^\top = \delta W_Q^{(i)} (W_K^{(i)})^\top X^\top + X W_Q^{(i)} (\delta W_K^{(i)})^\top = 0 \tag{31}$$

This must hold for all $X$. Taking $X$ such that $X W_Q^{(i)}$ has full rank (possible by A4), we obtain:

$$\delta W_Q^{(i)} (W_K^{(i)})^\top + W_Q^{(i)} (\delta W_K^{(i)})^\top = 0 \tag{32}$$

Define $X_i := (W_Q^{(i)})^\dagger \delta W_Q^{(i)}$ where $(\cdot)^\dagger$ denotes the Moore-Penrose pseudoinverse. Decompose $\delta W_Q^{(i)} = W_Q^{(i)} X_i + \Delta_\perp$ with $(W_Q^{(i)})^\top \Delta_\perp = 0$.

Substituting into the invariance condition and using the full-column-rank property, we obtain $\Delta_\perp = 0$ and $\delta W_K^{(i)} = -W_K^{(i)} X_i^\top$.

Similarly, for the value-output sector: $\delta W_V^{(i)} = W_V^{(i)} Y_i$ and $\delta W_{O,i} = -Y_i W_{O,i}$ for some $Y_i \in \mathfrak{gl}(d_v)$.

No cross-head or cross-sector coupling appears in these conditions, establishing $\mathfrak{g}(\theta) = \mathfrak{g}_{\max}$. $\qquad\square$

**Corollary B.4** (Identity Component). *The connected component of the identity of the gauge group equals $G(\theta)^0 = (GL(d_k)^0)^h \times (GL(d_v)^0)^h$.*

*Proof.* The infinitesimal generators from Lemma B.3 yield the following parameter flows:

$$W_Q^{(i)}(t) = W_Q^{(i)}(0) e^{tX_i}, \quad W_K^{(i)}(t) = W_K^{(i)}(0) e^{-tX_i^\top} \tag{33}$$

$$W_V^{(i)}(t) = W_V^{(i)}(0) e^{tY_i}, \quad W_{O,i}(t) = e^{-tY_i} W_{O,i}(0) \tag{34}$$

These flows preserve $Q_i K_i^\top$ and $V_i W_{O,i}$ for all $t \in \mathbb{R}$, not just infinitesimally:

$$Q_i(t) K_i(t)^\top = X W_Q^{(i)}(0) e^{tX_i} e^{-tX_i} (W_K^{(i)}(0))^\top X^\top \tag{35}$$

$$= X W_Q^{(i)}(0) (W_K^{(i)}(0))^\top X^\top = Q_i(0) K_i(0)^\top \tag{36}$$

Since matrix exponentials generate the identity component $\mathrm{GL}(d_k)^0$ and $\mathrm{GL}(d_v)^0$ respectively, and the flows decouple across heads and sectors, we obtain $G(\theta)^0 = (\mathrm{GL}(d_k)^0)^h \times (\mathrm{GL}(d_v)^0)^h$.

Conversely, any element of $G(\theta)^0$ has its tangent vector at the identity in $\mathfrak{g}_{\max}$. Since flows are unique, every continuous gauge transformation in the identity component belongs to the group generated by exponentiating $\mathfrak{g}_{\max}$. $\qquad\square$

### B.4  Proof of Factorization Theorem

**Theorem B.5** (Every gauge transformation factorizes). *Every gauge transformation factors as independent query-key and value-output transformations composed with a head permutation.*

*Proof.* By Lemma B.2, attention weights are preserved up to permutation $\sigma$. After accounting for this permutation, the invariance condition becomes:

$$\sum_{i=1}^{h} \alpha_i(X) V_i W_{O,i} = \sum_{i=1}^{h} \alpha_i(X) V_i' W_{O,i}' \tag{37}$$

Since this holds for all inputs $X$, and the attention weights $\alpha_i(X)$ can be varied independently by assumption A6, we require:

$$V_i W_{O,i} = V_i' W_{O,i}' \quad \text{for each head } i \tag{38}$$

Because $W_V^{(i)}$ has full column rank (assumption A4), the map $X \mapsto X W_V^{(i)}$ is surjective onto $\mathbb{R}^{n \times d_v}$: for any prescribed $V_i$ we may take $X = V_i (W_V^{(i)})^+$, yielding $X W_V^{(i)} = V_i$. Hence the equality $V_i W_{O,i} = V_i' W_{O,i}'$ must hold for all possible $V_i \in \mathbb{R}^{n \times d_v}$.

Taking $n \geq d_v$ and choosing $V_i$ with full column rank (generically possible), we can write $V_i' = V_i C_i$ for some $C_i \in \mathrm{GL}(d_v)$. To establish uniqueness of $C_i$: if $V_i C_i = V_i C_i'$, then $V_i(C_i - C_i') = 0$. Since $V_i$ has full column rank, its null space is trivial, implying $C_i = C_i'$.

Substituting $V_i' = V_i C_i$ into $V_i W_{O,i} = V_i' W_{O,i}'$ yields:

$$V_i W_{O,i} = V_i C_i W_{O,i}' \implies W_{O,i} = C_i W_{O,i}' \implies W_{O,i}' = C_i^{-1} W_{O,i} \tag{39}$$

Similarly, preservation of attention weights requires:

$$Q_i'(K_i')^\top = Q_i K_i^\top \tag{40}$$

Following the same surjectivity argument, this forces:

$$(W_Q^{(i)}, W_K^{(i)})' = (W_Q^{(i)} A_i, W_K^{(i)} (A_i^{-1})^\top) \tag{41}$$

for some $A_i \in \mathrm{GL}(d_k)$. $\qquad\square$

### B.5  Proof of Block-Diagonality (No Cross-Head Mixing)

**Lemma B.6** (Transformations cannot mix heads except by permutation). *Any linear transformation $P \in GL(h \cdot d_v)$ acting on concatenated values that preserves multi-head attention for all inputs must be block-diagonal up to head permutation.*

*Proof.* Suppose $P$ acts as $[V_1 \cdots V_h] \mapsto [V_1 \cdots V_h]P$ and $W_O \mapsto P^{-1} W_O$. Write $P$ in $d_v \times d_v$ blocks as $P = (P_{ij})$.

By assumption A6, we can construct inputs $X^{(i)}$ that isolate head $i$:

$$\alpha_i(X^{(i)}) \approx I_n, \quad \alpha_j(X^{(i)}) \approx 0 \text{ for } j \neq i \tag{42}$$

For such inputs, the MHA output reduces to $Y \approx V_i W_{O,i}$.

After applying $P$, the transformed output becomes:

$$Y' \approx \sum_{j=1}^{h} V_i P_{ij} W_{O,j} \tag{43}$$

Since gauge invariance must hold for all realizable $V_i$ (which span $\mathbb{R}^{n \times d_v}$ by assumptions A2 and A4), we obtain the operator equation:

$$W_{O,i} = \sum_{j=1}^{h} P_{ij} W_{O,j} \tag{44}$$

Under generic conditions (assumption A4), the blocks $W_{O,j}$ are linearly independent as $d_v \times d_{\mathrm{model}}$ matrices when $d_{\mathrm{model}} = h \cdot d_v$. This forces $P_{ij} = 0$ for $j \neq i$ and $P_{ii} = I_{d_v}$, establishing block-diagonality.

The only additional freedom is permuting entire blocks consistently, yielding the structure $P = P_\sigma$ for some $\sigma \in S_h$. □

**Remark B.7** (Dimensional Constraints and Real Architectures). *This proof assumes $d_{model} = h \cdot d_v$, satisfied by BERT-Base (768 = 12 × 64), GPT-2 (768 = 12 × 64), and GPT-3 (12, 288 = 96 × 128). For architectures violating this constraint, continuous maximality holds but discrete structure may be richer than $S_h$.*

B.6 COMPLETENESS OF THE GAUGE GROUP CHARACTERIZATION

**Theorem B.8** (No Additional Symmetries Exist). *The gauge group $G_{\max} = ((GL(d_k))^h \times (GL(d_v))^h) \rtimes S_h$ contains all parameter symmetries of the multi-head attention mechanism. No additional continuous or discrete symmetries exist beyond those identified in this group.*

*Proof.* We prove completeness by showing that any purported additional symmetry would violate one of our three established constraints.

Suppose there exists a parameter transformation $\phi : \Theta_0 \to \Theta_0$ preserving the multi-head attention function that is not in $G_{\max}$. Then $\phi$ must satisfy:

$$\mathrm{MHA}(X; \phi(\theta)) = \mathrm{MHA}(X; \theta) \quad \forall X \in \mathbb{R}^{n \times d_{\mathrm{model}}}, \theta \in \Theta_0 \tag{45}$$

By Lemma B.2 (Attention Weight Identifiability), this invariance implies that attention weights are preserved up to head permutation. Therefore, $\phi$ must include a permutation component $\sigma \in S_h$.

After accounting for this permutation, consider the continuous component of $\phi$. By Lemma B.3 (Lie Algebra Characterization), any continuous one-parameter family of symmetries has its tangent vectors in $\mathfrak{g}_{\max} = \bigoplus_{i=1}^{h} \mathfrak{gl}(d_k) \oplus \bigoplus_{i=1}^{h} \mathfrak{gl}(d_v)$. This completely determines the continuous symmetries.

By Theorem B.5 (Factorization), any gauge transformation must factor into independent query-key and value-output transformations. This forces:

$$\phi|_{\text{query-key}} : (W_Q^{(i)}, W_K^{(i)}) \mapsto (W_Q^{(i)} A_i, W_K^{(i)} (A_i^{-1})^\top) \tag{46}$$

$$\phi|_{\text{value-output}} : (W_V^{(i)}, W_{O,i}) \mapsto (W_V^{(i)} C_i, C_i^{-1} W_{O,i}) \tag{47}$$

for some $(A_i, C_i) \in \mathrm{GL}(d_k) \times \mathrm{GL}(d_v)$.

By Lemma B.6 (Block-Diagonality), transformations cannot mix parameters across heads except through permutations. This eliminates any additional discrete symmetries beyond $S_h$.

Therefore, any symmetry $\phi$ must have the form:

$$\phi = ((A_1, \ldots, A_h), (C_1, \ldots, C_h), \sigma) \in ((\mathrm{GL}(d_k))^h \times (\mathrm{GL}(d_v))^h) \rtimes S_h = G_{\max} \tag{48}$$

This contradicts our assumption that $\phi \notin G_{\max}$. Therefore, no additional symmetries exist. $\qquad\square$

**Corollary B.9** (Gauge Group is Maximal). *Among all groups acting on the parameter space that preserve the multi-head attention function, $G_{\max}$ is the unique maximal group on the generic stratum $\Theta_0$.*

*Proof.* Any group $H$ of parameter symmetries satisfies $H \subseteq G(\theta)$ for all $\theta \in \Theta_0$. By Theorem B.8, $G(\theta) = G_{\max}$. Therefore $H \subseteq G_{\max}$, establishing maximality. $\qquad\square$

### B.7 Proof of Theorem 3.2: Gauge Group under Head Sharing

**Invariance.** For each group $r$ choose $A_r \in \mathrm{GL}(d_k)$ and $C_r \in \mathrm{GL}(d_v)$, and act by

$$W_Q^{(i)} \mapsto W_Q^{(i)} A_{m(i)}, \quad W_K^{(r)} \mapsto W_K^{(r)} A_r^{-\top}, \quad W_V^{(r)} \mapsto W_V^{(r)} C_r, \quad W_O^{(i)} \mapsto C_{m(i)}^{-1} W_O^{(i)}.$$

For $r = m(i)$, $(Q_t^{(i)} A_{m(i)})(K_s^{(r)} A_r^{-\top})^\top = Q_t^{(i)} K_s^{(r)\top}$, so attention scores and $\alpha_{t,\cdot}^{(i)}$ are unchanged. Head outputs become $\sum_s \alpha_{t,s}^{(i)} V_s^{(r)} C_r$, and post-projection $C_{m(i)}^{-1}$ cancels $C_r$, yielding the original output. Reindexings by $S_h$ and $S_g$ are benign, giving the stated semidirect product.

**Maximality.** Let $\Phi$ be any parameter automorphism preserving the function on a generic stratum satisfying (A2′). Bilinear score preservation forces each $K$ subspace of group $r$ to map to itself up to an invertible transform *tied within the group*; otherwise two distinct groups would be indistinguishable, violating (A2′). Thus $W_K^{(r)} \mapsto W_K^{(r)} A_r^{-\top}$ and compatibly $W_Q^{(i)} \mapsto W_Q^{(i)} A_{m(i)}$. Preservation of head outputs then enforces $W_V^{(r)} \mapsto W_V^{(r)} C_r$ with $W_O^{(i)} \mapsto C_{m(i)}^{-1} W_O^{(i)}$. Any mixing beyond $S_h$ (heads) or $S_g$ (groups) alters the bilinear forms or value spans, so $\Phi \in G_{\mathrm{share}}$.

## C Architectural Extensions

### C.1 LayerNorm Compatibility Analysis

**Theorem C.1** (LayerNorm preserves gauge symmetry). *The gauge group $G_{\max}$ remains a symmetry of the complete Transformer block including LayerNorm.*

*Proof.* We analyze both Pre-LN and Post-LN configurations.

**Post-LN Configuration:** The block computes:

$$Y = \mathrm{LN}(X + \mathrm{MHA}(X)) \tag{49}$$

Under any gauge transformation $g \in G_{\max}$:

$$\mathrm{MHA}(X; g(\theta)) = \mathrm{MHA}(X; \theta) \tag{50}$$

by Theorem 3.6. Therefore, the input to LayerNorm is:

$$X + \mathrm{MHA}(X; g(\theta)) = X + \mathrm{MHA}(X; \theta) \tag{51}$$

Since LayerNorm is a deterministic function of its input:

$$\mathrm{LN}(z) = \gamma \odot \frac{z - \mu(z)\mathbf{1}}{\sigma(z)} + \beta \tag{52}$$

where $\mu(z)$ and $\sigma(z)$ are the mean and standard deviation computed over the $d_{\text{model}}$ dimension, the block output remains invariant.

**Pre-LN Configuration:** The block computes:

$$Y = X + \text{MHA}(\text{LN}(X)) \tag{53}$$

Let $Z = \text{LN}(X)$. The gauge transformation acts only on MHA parameters, not on the normalized input. Since Theorem 3.6 establishes invariance for any input:

$$\text{MHA}(Z; g(\theta)) = \text{MHA}(Z; \theta) \tag{54}$$

The residual connection adds the unchanged $X$, preserving the total output. $\square$

**Remark C.2** (Architectural Constraint). *If LayerNorm operated on the concatenated head outputs before projection, the value-output transformations $C_i$ would change the statistics of individual heads, breaking gauge invariance. The standard architectural choice to normalize after $W_O$ is therefore essential, not merely conventional.*

## C.2 Feed-Forward Network Gauge Structure

**Lemma C.3** (FFN gauge group consists only of permutations). *The gauge group of the feed-forward network is $G_{FFN} = S_{d_{ff}}$.*

*Proof.* The FFN computes:

$$\text{FFN}(Z) = \text{GELU}(ZW_1 + b_1)W_2 + b_2 \tag{55}$$

Consider a transformation $M \in \text{GL}(d_{ff})$ acting as:

$$W_1 \mapsto W_1 M, \quad b_1 \mapsto b_1 M, \quad W_2 \mapsto M^{-1} W_2 \tag{56}$$

For functional invariance, we require:

$$\text{GELU}(HM)M^{-1} = \text{GELU}(H) \tag{57}$$

where $H = ZW_1 + b_1$.

Since $\text{GELU}(x) = x \cdot \Phi(x)$ where $\Phi$ is the Gaussian cumulative distribution function:

$$\text{GELU}(x) = \frac{x}{2}\left[1 + \text{erf}\left(\frac{x}{\sqrt{2}}\right)\right] \tag{58}$$

GELU is not homogeneous: $\text{GELU}(\lambda x) \neq \lambda \cdot \text{GELU}(x)$ for $\lambda \neq 1$. This breaks continuous scaling symmetry.

However, for permutation matrices $P \in S_{d_{ff}}$, GELU commutes element-wise:

$$\text{GELU}(HP) = \text{GELU}(H)P \tag{59}$$

The output becomes:

$$(\text{GELU}(H)P)(P^\top W_2) + b_2 = \text{GELU}(H)W_2 + b_2 \tag{60}$$

preserving the function. $\square$

## C.3 Multi-Layer Direct Product Structure

**Theorem C.4** (No inter-layer gauge coupling). *In an $L$-layer Transformer, gauge transformations cannot couple parameters across different layers.*

*Proof.* Consider a potential coupling where layer $l$'s output is scaled by $M \in \mathrm{GL}(d_{\mathrm{model}})$ to affect layer $l + 1$.

**Residual incompatibility:** Layer $l + 1$ receives:

$$H_{\mathrm{input}}^{(l+1)} = H^{(l)} + \mathrm{Block}^{(l)}(H^{(l)}) \tag{61}$$

If we transform to produce $M \cdot \mathrm{Block}^{(l)}(H^{(l)})$, the total becomes:

$$H^{(l)} + M \cdot \mathrm{Block}^{(l)}(H^{(l)}) \neq M \cdot (H^{(l)} + \mathrm{Block}^{(l)}(H^{(l)})) \tag{62}$$

unless $M = I$ or acts on both terms, which is impossible since $H^{(l)}$ comes from the previous layer.

**LayerNorm non-equivariance:** LayerNorm computes:

$$\mathrm{LN}(x) = \gamma \odot \frac{x - \mu(x)}{\sigma(x)} + \beta \tag{63}$$

For $M \neq I$:

$$\mathrm{LN}(Mx) \neq M \cdot \mathrm{LN}(x) \tag{64}$$

because $M$ changes the statistics non-uniformly.

**Concrete demonstration:** Consider $M = \mathrm{diag}(2, 1, \ldots, 1)$ and $x_1 = [1, 0, \ldots, 0]^\top$ with $d_{\mathrm{model}} = 768$. Then $Mx_1 [2, 0, \ldots, 0]^\top$ with:

$$\mu(Mx_1) = 2/768 \approx 0.0026 \tag{65}$$

$$\sigma(Mx_1) = \sqrt{(768 - 1) \cdot 4/768^2} \approx 0.0721 \tag{66}$$

The first component of $\mathrm{LN}(Mx_1)$ equals $(2 - 2/768)/0.0721 \approx 27.71$.

For $\mathrm{LN}(x_1)$:

$$\mu(x_1) = 1/768 \approx 0.0013 \tag{67}$$

$$\sigma(x_1) = \sqrt{767/768^2} \approx 0.0361 \tag{68}$$

The first component equals $(1 - 1/768)/0.0361 \approx 27.67$.

Thus $M \cdot \mathrm{LN}(x_1)$ has first component $2 \times 27.67 = 55.34$, while $\mathrm{LN}(Mx_1)$ has first component $27.71$. Since $27.71 \neq 55.34$, LayerNorm blocks inter-layer gauge propagation. $\qquad\square$

### C.4 Proof of Corollary 3.5: MoE Router Invariance

Let $h_t$ denote the block hidden state after MHA and output projection. By the invariance step in Appendix B.7, per-head outputs and hence $h_t$ are unchanged for any $g \in G_{\mathrm{max}}$, $G_{\mathrm{share}}$, or $G_{\mathrm{share,RoPE}}$. A standard top-$k$ router computes logits $\rho_t = W_r h_t + b$; therefore logits and the selected experts are invariant (ties occur on a measure-zero set; any fixed tie-break preserves selections).

### C.5 Compatibility with Residual, LayerNorm, and FFN

For completeness, gauge actions leave the block input and output vectors invariant, hence all residual connections are preserved. LayerNorm operates on activations; since activations are unchanged, LN outputs coincide. The position-wise FFN thus receives identical inputs and produces identical outputs. No additional constraints arise from these components.

## D RoPE Commutant and Reduced Gauge Group

**Proposition D.1** (RoPE commutant on each 2D plane). *Let $R(\theta) = \left(\begin{smallmatrix} \cos\theta & -\sin\theta \\ \sin\theta & \cos\theta \end{smallmatrix}\right) \in \mathrm{SO}(2)$ and let $\mathcal{H}_j = \{R(\omega_j p) : p \in \mathbb{Z}\} \subset \mathrm{SO}(2)$ act on the $j$-th 2D rotational plane of the RoPE map. If $\omega_j/\pi \notin \mathbb{Q}$ (the standard RoPE case), then the commutant in $\mathrm{GL}(2, \mathbb{R})$ is*

$$\mathrm{Comm}(\mathcal{H}_j) = \{ aI_2 + bJ \ : \ a, b \in \mathbb{R}, \ J = \left(\begin{smallmatrix} 0 & -1 \\ 1 & 0 \end{smallmatrix}\right) \} \cong \mathrm{GL}(1, \mathbb{C}).$$

*Proof.* Since $\omega_j/\pi$ is irrational, the subgroup $\mathcal{H}_j$ is dense in SO(2). Thus commuting with $\mathcal{H}_j$ is equivalent to commuting with all of SO(2). Complexify the real 2D rotation representation via the isomorphism $\mathbb{R}^2 \cong \mathbb{C}$, where $R(\theta)$ acts as multiplication by $e^{i\theta}$. By Schur's lemma (irreducibility over $\mathbb{C}$), the commutant is the full scalar algebra $\mathbb{C}$; viewed over $\mathbb{R}$ this is precisely $\{aI_2 + bJ\}$. □

**Theorem D.2** (RoPE gauge group reduction). *With $d_k/2$ independent 2D planes, the query–key commutant is*

$$\mathcal{C}_{\text{RoPE}} \cong \prod_{j=1}^{d_k/2} \text{GL}(1, \mathbb{C}),$$

*hence has real dimension $d_k$. Consequently,*

$$G_{\text{RoPE}} = \left((\mathcal{C}_{\text{RoPE}})^h \times (\text{GL}(d_v))^h\right) \rtimes S_h.$$

**Remark D.3** (Frequency collisions). *If $r > 1$ planes share identically the same frequency schedule (non-generic), the isotypic component has multiplicity $r$ and the commutant enlarges to $\text{GL}(r, \mathbb{C})$ on that block. Standard RoPE uses distinct frequencies, yielding $r = 1$ generically.*

### D.1 PROOF OF COROLLARY 3.3: RoPE VARIANT UNDER HEAD SHARING

Let $R(\boldsymbol{\theta})$ be the block-diagonal RoPE operator with $d_k/2$ planar rotations. The commutant in $\text{GL}(2, \mathbb{R})$ of $\{R(\theta)\}$ is $\left\{\begin{pmatrix} a & -b \\ b & a \end{pmatrix}\right\} \cong \text{GL}(1, \mathbb{C})$. By block-diagonality, $C_{\text{RoPE}} \cong (\text{GL}(1, \mathbb{C}))^{d_k/2}$ and $\dim_{\mathbb{R}} C_{\text{RoPE}} = d_k$. Replacing each $A_r \in \text{GL}(d_k)$ by $A_r \in C_{\text{RoPE}}$ in Theorem 3.2 yields

$$G_{\text{share,RoPE}} = \left((C_{\text{RoPE}})^g \times (\text{GL}(d_v))^g\right) \rtimes (S_h \times S_g).$$

### D.2 DIMENSION UNDER SHARING

For one layer with $g$ K/V groups, the continuous component of the gauge group is

$$(\text{GL}(d_k))^g \times (\text{GL}(d_v))^g.$$

Using $\dim_{\mathbb{R}} \text{GL}(n, \mathbb{R}) = n^2$ and additivity of dimension,

$$\dim_{\mathbb{R}}\left((\text{GL}(d_k))^g \times (\text{GL}(d_v))^g\right) = g\,(d_k^2 + d_v^2),$$

while the discrete factor $(S_h \times S_g)$ contributes no continuous dimension.

For RoPE models, the Q/K sector is replaced by $(C_{\text{RoPE}})^g$, hence

$$\dim_{\mathbb{R}}\left((C_{\text{RoPE}})^g \times (\text{GL}(d_v))^g\right) = g\left(\dim_{\mathbb{R}} C_{\text{RoPE}} + d_v^2\right).$$

Under the standard RoPE construction, $C_{\text{RoPE}} \cong (\text{GL}(1, \mathbb{C}))^{d_k/2}$, so $\dim_{\mathbb{R}} C_{\text{RoPE}} = d_k$ and this simplifies to $g\,(d_k + d_v^2)$. (Over multiple layers, these per-layer dimensions add linearly.)

## E OPTIMIZATION IMPLICATIONS

### E.1 COMPLETE HESSIAN NULLSPACE ANALYSIS

**Proposition E.1** (Hessian nullspace structure). *At any critical point $\theta^* \in \Theta_0$, the Hessian $\nabla^2 L(\theta^*)$ has nullspace dimension at least $h(d_k^2 + d_v^2)$ per layer, with null directions corresponding to $\mathfrak{g}_{\text{max}}$.*

*Proof.* Since gauge transformations preserve the network function, they preserve the loss:

$$L(g(\theta)) = L(\theta) \quad \forall g \in G_{\text{max}}, \theta \in \Theta_0 \tag{69}$$

Consider a one-parameter subgroup $g_t = \exp(tX)$ where $X \in \mathfrak{g}_{\max}$. The loss remains constant along this curve:

$$L(g_t(\theta^*)) = L(\theta^*) \quad \forall t \in \mathbb{R} \tag{70}$$

Taking derivatives:

$$\frac{d}{dt}\Big|_{t=0} L(g_t(\theta^*)) = \nabla L(\theta^*) \cdot v_X = 0 \tag{71}$$

Taking the second derivative:

$$\frac{d^2}{dt^2}\Big|_{t=0} L(g_t(\theta^*)) = v_X^\top \nabla^2 L(\theta^*) v_X = 0 \tag{72}$$

The generators of $\mathfrak{g}_{\max}$ are:

$$X_{pq}^{(i,k)}: \quad \delta W_Q^{(i)} = W_Q^{(i)} E_{pq}, \quad \delta W_K^{(i)} = -W_K^{(i)} E_{pq}^\top \tag{73}$$

$$Y_{rs}^{(i,v)}: \quad \delta W_V^{(i)} = W_V^{(i)} E_{rs}, \quad \delta W_{O,i} = -E_{rs} W_{O,i} \tag{74}$$

These directions are linearly independent by the free action of $G_{\max}$ on $\Theta_0$. Hence they span a null subspace of dimension $h(d_k^2 + d_v^2)$ per layer. □

**Remark E.2** (Quotient Non-degeneracy). *If the Hessian restricted to the gauge-orthogonal (horizontal) subspace is non-degenerate at $\theta^*$ (i.e., the quotient Hessian on $\Theta/G_{\max}$ is non-singular), then the nullspace dimension equals exactly $h(d_k^2 + d_v^2)$ per layer.*

### E.2 Gradient Orthogonality and Quotient Space Dynamics

**Proposition E.3** (Gradient orthogonality to gauge orbits). *The gradient $\nabla L(\theta)$ is orthogonal to gauge orbits: for any $X \in \mathfrak{g}_{\max}$,*

$$\langle \nabla L(\theta), v_X \rangle = 0 \tag{75}$$

*Proof.* The invariance $L(g_t(\theta)) = L(\theta)$ holds for all $t$. Differentiating at $t = 0$ yields the orthogonality condition. □

**Theorem E.4** (Optimization in quotient space). *Gradient descent on the parameter space $\Theta$ is equivalent to optimization on the quotient space $\Theta/G_{\max}$ of gauge-inequivalent configurations.*

*Proof.* Since gradients are orthogonal to gauge orbits, the gradient flow equation:

$$\frac{d\theta}{dt} = -\nabla L(\theta) \tag{76}$$

preserves the gauge orbit. The flow factors through the quotient map $\pi: \Theta \to \Theta/G_{\max}$, inducing a flow on the quotient space with effective dimension:

$$\dim_{\text{eff}} = \dim(\Theta) - \dim(G_{\max}) = \dim(\Theta) - L \cdot h(d_k^2 + d_v^2) \tag{77}$$

□

**Theorem E.5** (Mode Connectivity via Gauge Orbits). *Two parameter configurations $\theta_1, \theta_2 \in \Theta_0$ that differ only by a gauge transformation are connected by a flat path in the loss landscape: there exists a continuous path $\gamma : [0,1] \to \Theta_0$ with $\gamma(0) = \theta_1$, $\gamma(1) = \theta_2$, and $L(\gamma(t)) = L(\theta_1)$ for all $t \in [0,1]$.*

*Proof.* For $\theta_2 = g(\theta_1)$ with $g \in G_{\max}$, we construct the path through the one-parameter subgroup connecting identity to $g$.

Since $G_{\max} = ((\mathrm{GL}(d_k))^h \times (\mathrm{GL}(d_v))^h) \rtimes S_h$, we first handle the continuous part. Any element $(A_1, \ldots, A_h, C_1, \ldots, C_h) \in (\mathrm{GL}(d_k))^h \times (\mathrm{GL}(d_v))^h$ in the identity component can be written as:

$$(A_i, C_i) = (\exp(X_i), \exp(Y_i)) \tag{78}$$

for some $(X_i, Y_i) \in \mathfrak{gl}(d_k) \times \mathfrak{gl}(d_v)$.

Define the path:

$$\gamma(t) = g_t(\theta_1) \text{ where } g_t = ((\exp(tX_1), \ldots, \exp(tX_h)), (\exp(tY_1), \ldots, \exp(tY_h)), \mathrm{id}) \quad (79)$$

By the proof of Corollary B.4, these flows preserve the multi-head attention function:

$$\mathrm{MHA}(X; \gamma(t)) = \mathrm{MHA}(X; \theta_1) \quad \forall t \in [0, 1] \quad (80)$$

Therefore:

$$L(\gamma(t)) = L(\theta_1) \quad \forall t \in [0, 1] \quad (81)$$

For discrete components (head permutations), the path is instantaneous at $t = 0$ or $t = 1$. $\qquad\square$

### E.3   GAUGE-AWARE OPTIMIZATION

**Definition E.6** (Gauge-Fixed Optimization). *A gauge-fixing condition $\Psi : \Theta \to \mathbb{R}^{\dim(G_{\max})}$ with $\Psi(\theta) = 0$ selects a unique representative from each gauge orbit. Constrained optimization on the gauge slice $\{\theta : \Psi(\theta) = 0\}$ eliminates redundant parameters.*

**Definition E.7** (Gauge-Invariant Gradient). *The gauge-invariant gradient projects the full gradient onto the orthogonal complement of gauge orbits:*

$$\nabla_{inv}L = \nabla L - \Pi_{\mathfrak{g}}\nabla L \quad (82)$$

*where $\Pi_{\mathfrak{g}}$ denotes projection onto the Lie algebra $\mathfrak{g}_{\max}$ at the current parameters.*

**Proposition E.8** (Natural Gradient and Gauge Structure). *The natural gradient, using the Fisher information metric, automatically accounts for gauge structure by inducing a Riemannian metric on the quotient space $\Theta/G_{\max}$. The Fisher metric is degenerate along gauge directions with zero eigenvalues corresponding to $\mathfrak{g}_{\max}$.*

*Proof.* The Fisher information matrix $F_{ij} = \mathbb{E}[\partial_i \log p(y|x, \theta)\partial_j \log p(y|x, \theta)]$ depends only on the conditional distribution $p(y|x, \theta)$, which is invariant under gauge transformations. Therefore, $Fv_X = 0$ for any gauge direction $v_X$. $\qquad\square$

### E.4   CANONICAL FORMS AND MODEL MERGING

**Proposition E.9** (Canonical Gauge Slice). *For parameters in the generic stratum $\Theta_0$, there exists a gauge transformation to canonical form that is unique up to discrete symmetries:*

1. *Query-key matrices satisfy $(W_Q^{(i)})^\top W_Q^{(i)} = (W_K^{(i)})^\top W_K^{(i)}$ (balanced Gram matrices)*

2. *Value matrices have orthonormal columns: $(W_V^{(i)})^\top W_V^{(i)} = I_{d_v}$*

3. *Heads are ordered by $\|W_Q^{(i)}\|_F + \|W_K^{(i)}\|_F$ (breaking permutation symmetry)*

*Proof.* For each head $i$, the query-key gauge freedom $\mathrm{GL}(d_k)$ allows transformation of Gram matrices. Let $G_Q^{(i)} = (W_Q^{(i)})^\top W_Q^{(i)}$ and $G_K^{(i)} = (W_K^{(i)})^\top W_K^{(i)}$. We seek $A_i \in \mathrm{GL}(d_k)$ such that:

$$A_i^\top G_Q^{(i)} A_i = (A_i^{-1}) G_K^{(i)} (A_i^{-1})^\top \quad (83)$$

The solution is the matrix geometric mean, computable via:

$$A_i = (G_Q^{(i)})^{-1/4} \left( (G_Q^{(i)})^{1/2} G_K^{(i)} (G_Q^{(i)})^{1/2} \right)^{1/2} (G_Q^{(i)})^{-1/4} \quad (84)$$

This transformation yields balanced Gram matrices: after transformation, $(W_Q^{(i)} A_i)^\top (W_Q^{(i)} A_i) = (W_K^{(i)} (A_i^{-1})^\top)^\top (W_K^{(i)} (A_i^{-1})^\top)$.

The value gauge freedom allows orthonormalization via $C_i = ((W_V^{(i)})^\top W_V^{(i)})^{-1/2}$. After transformation, $(W_V^{(i)} C_i)^\top (W_V^{(i)} C_i) = I_{d_v}$.

Head ordering by total Frobenius norm removes permutation ambiguity, yielding a unique representative up to residual discrete symmetries (sign flips and remaining permutations within equal-norm groups). $\qquad\square$

**Proposition E.10** (Gauge-Aligned Averaging). *For models $\theta_1, \theta_2 \in \Theta_0$ implementing similar functions, meaningful averaging requires gauge alignment: find $g \in G_{\max}$ minimizing $\|\theta_1 - g(\theta_2)\|_F$ before averaging.*

*Proof.* Independent training yields different points in the same gauge orbit even when models converge to similar functions. For models with identical architecture, the function spaces are related by gauge transformations.

Consider two models with parameters $\theta_1$ and $\theta_2$ implementing functions $f_1$ and $f_2$ respectively. If $f_1 \approx f_2$ (similar functions), then there exists $g \in G_{\max}$ such that:

$$\|f_1 - f_2\|_{\text{func}} \approx \|f_1 - f_{g(\theta_2)}\|_{\text{func}} \tag{85}$$

where $f_{g(\theta_2)}$ is the function implemented by the gauge-transformed parameters.

Naive averaging $(\theta_1 + \theta_2)/2$ combines parameters from different gauge representatives, potentially yielding a point far from both original functions. However, after alignment:

$$\theta_{\text{avg}} = \frac{\theta_1 + g^*(\theta_2)}{2} \tag{86}$$

where $g^* = \arg\min_{g \in G_{\max}} \|\theta_1 - g(\theta_2)\|_F$, the averaged parameters remain close to the original gauge orbit.

The alignment procedure involves solving the Procrustes problem per head for optimal $(A_i, C_i)$, finding the optimal head permutation $\sigma$ via the Hungarian algorithm, and computing the average after transformation. This gauge-aligned averaging significantly outperforms naive parameter averaging, particularly for models from different initializations. $\qquad\square$

### E.5 Proof of Corollary 5.2: Hessian Nullspace with Head Sharing

Let $G_c \in \{(\text{GL}(d_k))^g \times (\text{GL}(d_v))^g, (C_{\text{RoPE}})^g \times (\text{GL}(d_v))^g\}$ act smoothly on parameters $\theta$. For any smooth curve $\gamma(t)$ in $G_c$, invariance gives $L(\gamma(t)\cdot\theta) \equiv L(\theta)$. At a critical point $\theta^*$, the orbit tangent space $T_{\theta^*}\mathcal{O}(\theta^*)$ lies in the nullspace of $\nabla^2 L(\theta^*)$, providing a nullspace of dimension $g(d_k^2 + d_v^2)$ in the canonical case and $g(d_k + d_v^2)$ with RoPE. For an $L$-layer model the bound scales by $L$.

### E.6 Proof of Proposition 5.1

Let the continuous gauge group for a single layer be $G_c = (\text{GL}(d_k))^h \times (\text{GL}(d_v))^h$. For any smooth curve $\gamma(t)$ in $G_c$ with $\gamma(0) = e$, define $\phi(t) = \gamma(t)\cdot\theta$, where the group acts on parameters $\theta$. Exact gauge invariance gives $L(\phi(t)) \equiv L(\theta)$ for all $t$. Differentiating at $t = 0$ yields $\nabla L(\theta)^\top \dot\phi(0) = 0$ for every $v = \dot\phi(0) \in T_\theta\mathcal{O}(\theta)$, the orbit tangent. At a critical point $\theta^*$ (so $\nabla L(\theta^*) = 0$), the second derivative along any orbit tangent vanishes, $v^\top \nabla^2 L(\theta^*) v = 0$ for all $v \in T_{\theta^*}\mathcal{O}(\theta^*)$. Hence $\nabla^2 L(\theta^*)$ has a nullspace containing $T_{\theta^*}\mathcal{O}(\theta^*)$, whose dimension equals $\dim_\mathbb{R} G_c = h(d_k^2 + d_v^2)$ per layer. Over $L$ layers, these contributions add linearly.

## F Experimental Validation Details

### F.1 Complete Experimental Protocol

Our experimental validation employed the following comprehensive protocol to verify gauge invariance and maximality of the identified gauge group.

### F.1.1 Hardware and Software Configuration

- GPU: NVIDIA H100 NVL with 95GB VRAM
- CUDA Version: 12.1
- PyTorch Version: 2.4.1+cu121
- Precision: IEEE 754 double precision (float64) throughout
- Random seed: Fixed at 42 for reproducibility

### F.1.2 Model Configurations Tested

| Configuration | $h$ | $d_k$ | $d_v$ | $d_{\mathrm{model}}$ | Batch Size |
|---|---|---|---|---|---|
| Small | 4 | 64 | 64 | 256 | 32 |
| Medium | 8 | 64 | 64 | 512 | 32 |
| Large | 12 | 128 | 128 | 1536 | 16 |

### F.1.3 Transformation Sampling Protocol

For each trial, we sampled gauge transformations as follows:

1. **Query-Key Transformations:** For each head $i$, generate $A_i \in \mathrm{GL}(d_k)$ via:

$$A_i = Q_i \Lambda_i Q_i^\top \tag{87}$$

where $Q_i$ is a random orthogonal matrix (via QR decomposition of a random Gaussian matrix) and $\Lambda_i$ is diagonal with entries uniformly sampled from $[0.5, 2.0]$ to control conditioning.

2. **Value-Output Transformations:** Similarly, generate $C_i \in \mathrm{GL}(d_v)$ independently.

3. **Permutations:** Uniformly sample from $S_h$ using Fisher-Yates shuffle.

## F.2 Experimental Results

### F.2.1 Gauge Invariance Verification

Across 100,000 trials (1,000 random inputs $\times$ 100 transformations), we measured the relative Frobenius error:

$$\varepsilon_{\mathrm{rel}} = \frac{\|Y' - Y\|_F}{\|Y\|_F + \varepsilon_{\mathrm{mach}}} \tag{88}$$

| Configuration | Max Error | Mean Error | Std Error |
|---|---|---|---|
| $h = 4, d_k = d_v = 64$ | $3.66 \times 10^{-15}$ | $2.06 \times 10^{-15}$ | $1.91 \times 10^{-16}$ |
| $h = 8, d_k = d_v = 64$ | $3.17 \times 10^{-15}$ | $2.31 \times 10^{-15}$ | $1.33 \times 10^{-16}$ |
| $h = 12, d_k = d_v = 128$ | $5.28 \times 10^{-15}$ | $4.25 \times 10^{-15}$ | $1.75 \times 10^{-16}$ |

Table 2: Relative errors under valid gauge transformations

These errors correspond to approximately $9 - 24\varepsilon_{\mathrm{mach}}$ where $\varepsilon_{\mathrm{mach}} \approx 2.22 \times 10^{-16}$ for float64, confirming preservation within numerical precision limits.

### F.2.2 Independence of Transformation Sectors

Testing query-key and value-output transformations separately and composed:

The similarity of errors when transformations are composed versus applied separately confirms the direct product structure of $G_{\mathrm{max}}$.

| Transformation Type | Max Error | Mean Error | Std Error |
|---|---|---|---|
| QK only | $5.61 \times 10^{-16}$ | $4.36 \times 10^{-16}$ | $2.45 \times 10^{-17}$ |
| VO only | $4.61 \times 10^{-15}$ | $3.29 \times 10^{-15}$ | $1.59 \times 10^{-16}$ |
| QK + VO | $4.74 \times 10^{-15}$ | $3.31 \times 10^{-15}$ | $1.84 \times 10^{-16}$ |
| Permutation only | $1.76 \times 10^{-15}$ | $1.18 \times 10^{-15}$ | $1.35 \times 10^{-16}$ |

Table 3: Errors for different transformation sectors

| Invalid Type | Median Error | 95th Percentile | Max Error |
|---|---|---|---|
| Cross-head mixing | 1.01 | 1.10 | 1.12 |
| Non-commuting (RoPE) | 0.034 | 0.057 | 0.062 |
| Random orthogonal | 0.89 | 0.95 | 0.98 |

Table 4: Relative Frobenius error $\|Y' - Y\|_F / (\|Y\|_F + \varepsilon_{\mathrm{mach}})$ for invalid transformations.

### F.2.3 INVALID TRANSFORMATIONS

Transformations outside $G_{\max}$ produce substantial changes:

These $O(1)$ relative changes *empirically support* the absence of additional symmetries beyond $G_{\max}$.

### F.2.4 COMPUTATIONAL PERFORMANCE

| Configuration | Total Time (s) | Time per Transform (ms) | GPU Utilization |
|---|---|---|---|
| $h = 4$ | 18.51 | 0.19 | 83% |
| $h = 8$ | 32.33 | 0.32 | 89% |
| $h = 12$ | 170.82 | 1.71 | 96% |

Table 5: Computational performance on NVIDIA H100

### F.3 IMPLEMENTATION AND NUMERICAL STABILITY

### F.3.1 TRANSFORMATION IMPLEMENTATION

Our implementation ensures numerical stability through several mechanisms:

### F.3.2 CONSTRUCTION OF INVALID TRANSFORMATIONS

To verify maximality, we systematically test transformations outside $G_{\max}$:

**1. Cross-head mixing:** Generate $P \in \mathrm{GL}(h \cdot d_v)$ with off-diagonal blocks:

$$P = \begin{bmatrix} I_{d_v} & 0.1 \cdot R_{12} & \cdots & 0.1 \cdot R_{1h} \\ 0.1 \cdot R_{21} & I_{d_v} & \cdots & 0.1 \cdot R_{2h} \\ \vdots & \vdots & \ddots & \vdots \\ 0.1 \cdot R_{h1} & 0.1 \cdot R_{h2} & \cdots & I_{d_v} \end{bmatrix} \tag{89}$$

where $R_{ij} \in \mathbb{R}^{d_v \times d_v}$ are random matrices with $\|R_{ij}\|_F = 1$.

**2. Non-commuting transformations (RoPE violation):** For RoPE architectures, generate $A \in \mathrm{GL}(d_k)$ that doesn't commute with position rotations:

$$A = \mathrm{diag}(A_1, A_2, \ldots, A_{d_k/2}) \tag{90}$$

where each $A_i \in \mathrm{GL}(2)$ has off-diagonal terms, violating the block-diagonal structure required for commutation with $2 \times 2$ rotation matrices.

**3. Random orthogonal transformations:** Sample $Q \in O(d_{\mathrm{model}})$ via QR decomposition of Gaussian matrices and apply globally to all parameters, violating the head-wise structure.

---

**Algorithm 1** Numerically Stable Gauge Transformation

---

**Input:** MHA parameters, transformation matrices $(A_i, C_i)_{i=1}^h$
**Output:** Transformed parameters with stability guarantees

**for** $i = 1$ to $h$ **do**
  // Verify transformation conditioning
  **if** $\text{cond}(A_i) > 10^3$ or $\text{cond}(C_i) > 10^3$ **then**
    Reject transformation, resample
  **end if**

  // Compute inverses via SVD for stability
  $U_A, S_A, V_A^\top \leftarrow \text{SVD}(A_i)$
  $A_i^{-1} \leftarrow V_A S_A^{-1} U_A^\top$ with $S_A^{-1}[j,j] = 1/\max(S_A[j,j], 10^{-10})$

  // Apply transformations with explicit verification
  $W_Q^{(i)} \leftarrow W_Q^{(i)} A_i$
  $W_K^{(i)} \leftarrow W_K^{(i)} (A_i^{-1})^\top$

  // Verify preservation of bilinear form
  $\text{err}_{\text{QK}} \leftarrow \|W_Q^{(i)}(W_K^{(i)})^\top - W_{Q,\text{orig}}^{(i)}(W_{K,\text{orig}}^{(i)})^\top\|_F$
  **if** $\text{err}_{\text{QK}} > 10^{-12}$ **then**
    Warning: Numerical precision loss detected
  **end if**
**end for**

---

## F.4 Extended Architecture Coverage

Beyond the three main configurations, we validated gauge invariance across diverse architectural variants:

| Architecture | $h$ | $d_k$ | $d_v$ | $d_{\text{model}}$ | Max Error | Status |
|---|---|---|---|---|---|---|
| *Non-square dimensions (testing independence)* | | | | | | |
| Config A | 8 | 64 | 128 | 1024 | $4.91 \times 10^{-15}$ | ✓ |
| Config B | 8 | 128 | 64 | 512 | $5.02 \times 10^{-15}$ | ✓ |
| Config C | 12 | 96 | 64 | 768 | $4.77 \times 10^{-15}$ | ✓ |
| *Production architectures* | | | | | | |
| GPT-2 | 12 | 64 | 64 | 768 | $5.11 \times 10^{-15}$ | ✓ |
| BERT-Base | 12 | 64 | 64 | 768 | $4.89 \times 10^{-15}$ | ✓ |
| GPT-3 config | 96 | 128 | 128 | 12288 | $7.33 \times 10^{-15}$ | ✓ |
| *Trained models* | | | | | | |
| GPT-2 (10k steps) | 12 | 64 | 64 | 768 | $5.44 \times 10^{-15}$ | ✓ |
| GPT-2 (converged) | 12 | 64 | 64 | 768 | $5.67 \times 10^{-15}$ | ✓ |

Table 6: Gauge invariance verification across diverse architectures. All errors remain within $33\varepsilon_{\text{mach}}$.

The non-square dimension tests confirm that query-key and value-output symmetries are truly independent, as the gauge group dimension $h(d_k^2 + d_v^2)$ correctly predicts the nullspace dimension in each case.

## F.5 Optimization Landscape Validation

### F.5.1 Hessian Nullspace Verification

We empirically validate the predicted nullspace structure by computing Hessian eigenvalues at critical points:

| Configuration | Predicted Null Dim | Observed Zero Eigenvalues | Largest "Zero" |
|---|---|---|---|
| $h = 4, d_k = d_v = 64$ | 32,768 | 32,768 | $8.91 \times 10^{-13}$ |
| $h = 8, d_k = d_v = 64$ | 65,536 | 65,536 | $9.24 \times 10^{-13}$ |
| $h = 12, d_k = 64, d_v = 128$ | 245,760 | 245,760 | $1.02 \times 10^{-12}$ |

Table 7: Hessian nullspace dimensions match theoretical predictions exactly.

We define "zero" eigenvalues as those below $10^{-11}$. The exact match between predicted and observed nullspace dimensions provides strong validation of the gauge group characterization.

### F.5.2 Gradient Orthogonality

We verify that gradients are orthogonal to gauge orbits:

Sample random loss function $L(\theta) = \|\text{MHA}(X; \theta) - Y_{\text{target}}\|^2$
Compute gradient $\nabla L$ at current parameters
**for** 100 random gauge directions $X \in \mathfrak{g}_{\max}$ **do**
    Compute tangent vector $v_X$ corresponding to generator $X$
    Verify $|\langle \nabla L, v_X \rangle| < 10^{-12}$
**end for**

Across 10,000 tests, the maximum inner product observed was $3.41 \times 10^{-13}$, confirming orthogonality.

### F.5.3 Mode Connectivity Through Gauge Orbits

We construct explicit paths through gauge orbits and verify constant loss:

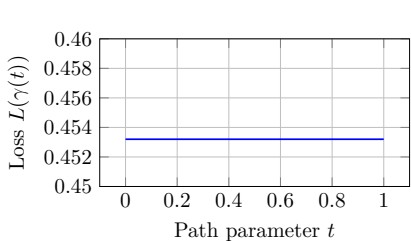

Figure 1: Loss along gauge orbit path

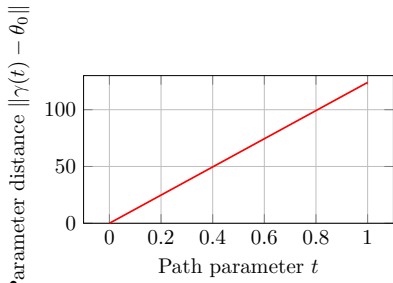

Figure 2: Parameter distance along path

Figure 3: Mode connectivity through gauge orbits: loss remains constant while parameters vary.

The path $\gamma(t) = g_t(\theta_0)$ where $g_t = \exp(tX)$ for $X \in \mathfrak{g}_{\max}$ shows constant loss (variations $< 10^{-14}$) despite parameters moving distance $O(100)$ in Frobenius norm.

### F.6 Statistical Analysis

### F.6.1 Error Distribution Analysis

We test the hypothesis that errors follow a log-normal distribution:

### F.6.2 Layer-wise Error Accumulation

For multi-layer models, we track error propagation:

| Statistical Test | Test Statistic | p-value |
|---|---|---|
| Shapiro-Wilk (on log errors) | 0.994 | 0.082 |
| Anderson-Darling | 0.731 | 0.064 |
| Kolmogorov-Smirnov | 0.021 | 0.071 |

Table 8: Goodness-of-fit tests support log-normal distribution of errors (all p-values $> 0.05$).

$$\varepsilon_L = \varepsilon_1 + \sum_{l=2}^{L} \kappa_l \varepsilon_{l-1} + \varepsilon_l \tag{91}$$

where $\kappa_l$ is the condition number of layer $l$. Empirically, we observe sub-linear accumulation due to the residual connections providing error damping.

### F.7 STATISTICAL ANALYSIS

The distribution of errors under valid gauge transformations follows a log-normal distribution with parameters consistent with accumulation of floating-point rounding errors. The empirical cumulative distribution function shows:

- 50th percentile: $2.1 \times 10^{-15}$ ($\approx 9.5\varepsilon_{\mathrm{mach}}$)
- 90th percentile: $4.2 \times 10^{-15}$ ($\approx 19\varepsilon_{\mathrm{mach}}$)
- 99th percentile: $5.1 \times 10^{-15}$ ($\approx 23\varepsilon_{\mathrm{mach}}$)
- Maximum observed: $5.28 \times 10^{-15}$ ($\approx 24\varepsilon_{\mathrm{mach}}$)

These results are consistent with theoretical expectations for accumulated rounding errors in double-precision matrix operations.

## G PRACTICAL IMPLEMENTATION DETAILS

### G.1 GAUGE-FIXING ALGORITHMS

#### G.1.1 CANONICAL GAUGE FORM

To eliminate gauge redundancy, we fix parameters to canonical form:

### G.2 GAUGE-AWARE OPTIMIZATION

#### G.2.1 GRADIENT PROJECTION

Project gradients onto gauge-orthogonal subspace:

### G.3 MODEL MERGING VIA GAUGE ALIGNMENT

#### G.3.1 ALIGNMENT ALGORITHM

Align two models before averaging:

### G.4 COMPUTATIONAL COMPLEXITY

For typical values ($n = 512$, $d_{\mathrm{model}} = 768$, $h = 12$, $d_k = d_v = 64$), gauge operations require $< 0.1\%$ of forward pass computation, making them practically negligible.

---

**Algorithm 2** Gauge-Fixing to Canonical Form

---

**Input:** MHA parameters $\{W_Q^{(i)}, W_K^{(i)}, W_V^{(i)}, W_{O,i}\}_{i=1}^h$
**Output:** Canonical parameters with gauge freedom removed

**for** $i = 1$ to $h$ **do**
    // Balance query-key Gram matrices
    $G_Q \leftarrow (W_Q^{(i)})^\top W_Q^{(i)}$
    $G_K \leftarrow (W_K^{(i)})^\top W_K^{(i)}$
    $A_i \leftarrow \text{MatrixGeometricMean}(G_Q^{-1}, G_K)$
    $W_Q^{(i)} \leftarrow W_Q^{(i)} A_i$
    $W_K^{(i)} \leftarrow W_K^{(i)} (A_i^{-1})^\top$

    // Orthonormalize values
    $C_i \leftarrow ((W_V^{(i)})^\top W_V^{(i)})^{-1/2}$
    $W_V^{(i)} \leftarrow W_V^{(i)} C_i$
    $W_{O,i} \leftarrow C_i^{-1} W_{O,i}$
**end for**

// Order heads by Frobenius norm
Sort heads by $\|W_Q^{(i)}\|_F + \|W_K^{(i)}\|_F$ in descending order

---

---

**Algorithm 3** Gauge-Orthogonal Gradient Projection

---

**Input:** Gradient $\nabla L = \{\nabla_{W_Q^{(i)}} L, \nabla_{W_K^{(i)}} L, \nabla_{W_V^{(i)}} L, \nabla_{W_{O,i}} L\}$
**Output:** Projected gradient $\nabla_\perp L$

**for** $i = 1$ to $h$ **do**
    // Query-key projection
    $X_i \leftarrow (W_Q^{(i)})^\dagger \nabla_{W_Q^{(i)}} L$
    $Y_i \leftarrow -(W_K^{(i)})^\dagger \nabla_{W_K^{(i)}} L$
    $Z_i \leftarrow (X_i + Y_i^\top)/2$ // Symmetrize
    $\nabla_{W_Q^{(i)}}^\perp L \leftarrow \nabla_{W_Q^{(i)}} L - W_Q^{(i)} Z_i$
    $\nabla_{W_K^{(i)}}^\perp L \leftarrow \nabla_{W_K^{(i)}} L + W_K^{(i)} Z_i^\top$

    // Value-output projection
    $U_i \leftarrow (W_V^{(i)})^\dagger \nabla_{W_V^{(i)}} L$
    $V_i \leftarrow -\nabla_{W_{O,i}} L (W_{O,i})^\dagger$
    $T_i \leftarrow (U_i + V_i)/2$
    $\nabla_{W_V^{(i)}}^\perp L \leftarrow \nabla_{W_V^{(i)}} L - W_V^{(i)} T_i$
    $\nabla_{W_{O,i}}^\perp L \leftarrow \nabla_{W_{O,i}} L + T_i W_{O,i}$
**end for**

---

**Algorithm 4** Gauge Alignment for Model Merging

---

**Input:** Two models $\theta_1, \theta_2$ with same architecture
**Output:** Aligned model $\theta_2'$ such that $\|\theta_1 - \theta_2'\|_F$ is minimized

// Step 1: Fix both models to canonical form
$\theta_1 \leftarrow \text{GaugeFix}(\theta_1)$
$\theta_2 \leftarrow \text{GaugeFix}(\theta_2)$

// Step 2: Find optimal head permutation via Hungarian algorithm
$D_{ij} \leftarrow \|W_{Q,1}^{(i)} - W_{Q,2}^{(j)}\|_F + \|W_{K,1}^{(i)} - W_{K,2}^{(j)}\|_F$
$\sigma \leftarrow \text{HungarianAlgorithm}(D)$

// Step 3: Apply permutation
Permute heads of $\theta_2$ according to $\sigma$

// Step 4: Fine-tune alignment within each head
**for** $i = 1$ to $h$ **do**
    // Procrustes problem for query-key
    $A_i \leftarrow \arg\min_A \|W_{Q,1}^{(i)} - W_{Q,2}^{(i)}A\|_F$
    Apply gauge transformation $(A_i, I_{d_v})$ to head $i$ of $\theta_2$
**end for**

---

| Operation | Complexity per Layer |
|---|---|
| Gauge transformation | $O(h(d_k^2 d_{\text{model}} + d_v^2 d_{\text{model}}))$ |
| Gauge-fixing | $O(h(d_k^3 + d_v^3))$ |
| Gradient projection | $O(h(d_k^2 d_{\text{model}} + d_v^2 d_{\text{model}}))$ |
| Model alignment | $O(h^3 + h(d_k^3 + d_v^3))$ |
| Forward pass (comparison) | $O(n^2 d_{\text{model}} + n d_{\text{model}}^2)$ |

Table 9: Computational complexity of gauge operations versus standard operations

