# OpenReview forum: "Maximal Gauge Symmetry in Transformer Architectures"
_ICLR.cc/2026/Conference — ICLR 2026 Conference Withdrawn Submission_

### Official Review · Reviewer_6gm3 · 2025-10-31

**Soundness:** 3
**Presentation:** 2
**Contribution:** 2
**Rating:** 4
**Confidence:** 5

**Summary:**

This paper discusses the symmetry properties of Transformers and defines their maximal gauge symmetry under various settings, including cases with or without positional encoding and with or without head sharing.

**Strengths:**

I believe the theoretical contribution is the strongest aspect of the paper.

- The authors define the group that captures all symmetries of the Transformer, encompassing both head permutations and the actions induced by the general linear group within each head. They also provide proofs demonstrating that this group accounts for all possible symmetries. While several assumptions are introduced for the sake of the proofs, these only exclude a negligible subset of the parameter space—specifically, sets of measure zero. I find this both necessary and consistent with prior works in the literature on the symmetry of neural architectures.

- The authors further extend their analysis to various Transformer variants, including those with rotary positional encodings (RoPE), head sharing, and multiple stacked Transformer layers.

**Weaknesses:**

While I appreciate the theoretical focus of the paper, I believe there are critical issues in the proofs.

- **On Lemma B.2 (attention weights preserved up to permutation).** This lemma underpins several main results, yet the first step of its proof, specifically Eq. (29), is problematic. If an input $X$ satisfies the first condition in Eq. (29), then $n$ must be sufficiently small (see lines 575–583). In contrast, the second condition seems to require $n$ to be sufficiently large: since each row of an attention matrix sums to 1, the Frobenius norm is bounded below by a positive constant for fixed $n$. These requirements appear incompatible. Although the lemma’s statement may be intuitively plausible, the given argument does not appear correct. More broadly, the core difficulty in analyzing multi-head attention symmetry is handling head permutations and the per-head GL actions simultaneously; this gap suggests the proof for the “original attention” case is unsound.

- **On RoPE.** The extension to rotary positional encodings is vague and, in my view, mis-specified. RoPE alters the attention structure, contrary to Eq. (4). In the commonly used formulation (as in the original RoPE paper), one should have $Q_i^{\text{RoPE}} = Q_iR_{\text{pos}}^i$, as in the original RoPE paper, that is also the variant that is widely used. The author missed the power $R_{\text{pos}}^i$, which canceled out the main differiant between usual PEs and RoPE, and the difficulty dissapear.

- **On LayerNorm and residual connections.** The analysis here is, in my opinion, straightforward and largely routine; nonetheless, I agree it is a contribution to completeness.

- **On multi-layer composition (Section 4).** The claim *"Since MHA and FFN operate on disjoint parameters and residual connections prevent coupling, their gauge groups combine as a direct product: $G_{\text{Block}} = G_{\text{MHA}} × G_{\text{FFN}}$."* seems overreaching. In general, for a composition $f_\theta \circ g_\gamma = f_{\theta'} \circ g_{\gamma'}$, knowing the symmetries of $f$ and $g$ separately does not determine the symmetry of $f \circ g$, particularly in the presence of residual pathways that can induce nontrivial interactions at the function level. I did read both the proof sketch and the detailed argument; this nuance appears to be overlooked, which weakens the multi-layer symmetry claim.

There are several minor errors in the presentation of the theoretical section, but they appear to be mostly typographical or easily fixable. Therefore, I only highlight the most significant issue.

Regarding the experimental aspect of the paper, it is unclear what specific applications the authors aim to address. The writing in this section is rather vague, and since no code has been submitted, the experimental results cannot be verified for correctness or reproducibility. As a result, it is difficult to assess what was actually implemented or demonstrated.

**Questions:**

Please address the issues I raised in the Weaknesses section concerning the theoretical analysis.

Regarding the experimental section, a concise summary of what was attempted would be sufficient. As I am not deeply familiar with the specific application domain, I only have one question about the GPT-3 experiments: since the paper focuses solely on model parameters, I am unsure how the authors were able to access the actual parameters of GPT-3, as they are not publicly available. My apologies if I have misunderstood this point.

---

Overall, I understand that the main focus of the paper lies in the theoretical contribution, so I do not weigh the limitations of the experimental part too heavily. However, despite appreciating the importance of the architectural symmetry problem the authors address, the presence of several significant errors in the theoretical analysis makes me hesitant to assign a high score.

---

### Official Review · Reviewer_UjbG · 2025-10-31

**Soundness:** 3
**Presentation:** 2
**Contribution:** 2
**Rating:** 6
**Confidence:** 4

**Summary:**

The paper looks into the gauge symmetry of Transformers and how it can be applied to things like understanding parameter redundancy, compressing models through gauge-fixing, improving optimization, merging or averaging models, and designing better architectures.

**Strengths:**

1. The paper is well-organized and clearly written.

2. The results on symmetry appear correct and are relatively straightforward to verify. The more challenging part lies in proving that the defined group action fully characterizes all symmetries, which the authors have addressed. The corresponding proofs are provided in the Appendix, though they involve mathematical tools outside my expertise, so I cannot confidently assess their correctness. It is commendable that the authors also extend their analysis to various types of attention mechanisms.

3. The experiments cover a broad range of applications.

**Weaknesses:**

1. The discussion of related work is somewhat incomplete and too brief. This raises the question of whether there are existing studies in the literature that have addressed gauge symmetry in Transformers or other architectures. The authors should expand this section to better position their work in relation to prior research.

2. No code submission was provided.

3. Appendix F and Appendix G are written quite carelessly: the tables and pseudo code are disorganized, lacking references or any clear explanations, making them difficult to follow. Appendix F does not clearly describe several important aspects of the experiments (such as dataset generation, model setup, and experimental design). In Appendix G, several algorithms are directly borrowed from previous papers, with only minor modifications to the symmetric group component to fit the theory developed in this work-for example, the Gauge Alignment for Model Merging method.

4. The paper only conducts experiments to evaluate output errors when applying a random Gauge Symmetry. This experiment is quite simple and does not provide much meaningful insight. Although Section 7 lists several potential applications of symmetry groups, the authors do not implement any specific application in practice. Moreover, most of the applications mentioned in Section 7 are standard use cases of symmetry groups, and the authors merely replace the symmetric group with a Gauge Symmetry to discuss its applicability.

**Questions:**

1. In line 357, the paper states: “Computational time scales linearly with model complexity, requiring approximately 170 seconds for the most complex configuration on an NVIDIA H100 GPU.” However, there are no experiments in the paper that appear to involve model training, so it is unclear what the GPU was actually used for in this context.

2. In Appendix F.5.2 about Gradient Orthogonality, it is not specified which dataset the model was trained on or what configuration was used. Was the model trained on any data, or was it simply randomly initialized? Additionally, it would be helpful to clarify which type of positional encoding was used for the Multi-Head Attention (MHA) in this experiment.

3. In Appendix F.5.3 about Mode Connectivity Through Gauge Orbits, the experimental setup is not clearly described. For instance, the paper does not indicate which dataset was used or how the experiment was designed. Furthermore, since the paper mentions that all experiments were conducted with a single random seed, it is unclear how two distinct models were obtained to evaluate linear mode connectivity.

4. Please clearly describe how your experiments were designed. For example, which datasets were used, how the models were generated (randomly initialized or trained), and what configurations were applied in each case.

---

### Official Review · Reviewer_Noub · 2025-10-31

**Soundness:** 3
**Presentation:** 2
**Contribution:** 3
**Rating:** 6
**Confidence:** 2

**Summary:**

The paper analyzes the gauge symmetry of Transformers and defines the maximal symmetry group under several scenarios, including those with RoPE, head sharing, MoE, and multilayer settings, with applications.

**Strengths:**

- All theoretical results are accompanied by proofs. The paper is mathematically intensive, and while I did not verify the proofs in detail, the results appear logically sound and readable.

- The analysis covers all key components of the Transformer, including attention, residual connections, layer normalization, and feedforward layers. The paper also discusses the case of multi-layer Transformers.

- The experimental section is diverse and explores a variety of settings.

**Weaknesses:**

- The presented results feel somewhat trivial. I’m not sure if this type of problem has been studied before for other deep learning models, since the introduction doesn’t provide enough background for readers who aren’t already familiar with the topic.

- The experimental section is quite brief and not very informative. It’s unclear what the authors are actually trying to demonstrate with the symmetry they defined. For the mode connectivity experiment, there are existing works on this topic—mainly for standard MLPs—where we usually see plots showing zero loss connectivity. I’m not sure whether the authors are checking mode connectivity for Transformer models here or something else.

**Questions:**

- Please expand the context and clearly state the research question, why gauge symmetry matters for Transformers, and how your results differ from prior symmetry analyses in deep learning.

- Please include a short subsection or paragraph that explains—at a high level—what the proposed symmetry results mean, why they are useful (e.g., for redundancy, compression, optimization, merging), and how readers from different backgrounds should interpret them.

- For each experiment, please specify the objective, datasets, model configuration (including positional encodings), training or initialization details, metrics, and the exact procedures used. State what each experiment is meant to demonstrate about the theory.

---

### Official Review · Reviewer_FB5i · 2025-10-31

**Soundness:** 3
**Presentation:** 2
**Contribution:** 2
**Rating:** 4
**Confidence:** 4

**Summary:**

The paper looks at the gauge symmetry of Transformers and defines the maximal symmetry group under different setups — like using or not using positional encoding, having or not having head sharing, and so on.

**Strengths:**

The paper is self-contained, heavy in mathematics, and a bit hard to follow, although it does have some efforts to sketch out the main idea. Results on the symmetry of Transformers are nice and have proofs. To the best of my knowledge, except for the original multihead attentions, the maximal symmetry for other cases, such as attention with RoPE, is novel.

**Weaknesses:**

- The paper should include a broader discussion of prior work on parameter symmetry and functional equivalence. Several important references are missing, such as [1] and [2]. It would also be helpful to discuss more about how symmetry has been used in applications like model merging or linear mode connectivity.

[1] Johanni Brea et al., Weight-space symmetry in deep networks gives rise to permutation saddles, connected by equal-loss valleys across the loss landscape.

[2] Phuong Bui Thi Mai and Christoph Lampert, Functional vs. parametric equivalence of ReLU networks.

- The assumptions (A1–A6) should be explained in more detail — specifically, why they are introduced and how each one influences the proofs.

- The formulation of Attention + RoPE does not seem consistent with the original RoPE paper [3]. In particular, the rotary matrix $R$ should be taken to the corresponding power, as defined in the original formulation.

[3] Jianlin Su et al., RoFormer: Enhanced Transformer with Rotary Position Embedding.

- The proof concerning stacking Transformer layers feels incomplete. Analyzing the symmetry of stacked layers is non-trivial, and it seems that some of these challenges may have been skipped in the current derivation.

- The experimental section is rather confusing, it is not clear how the proposed symmetry is applied in each task or during optimization.

**Questions:**

**Q1** Please clarify the motivation for introducing assumptions A1–A6. Why are these assumptions necessary, and how do they influence the validity or scope of the proofs?

**Q2** The formulation of Attention + RoPE in the paper seems to differ from that in the original RoPE paper [3]. Can the author clarify this point

**Q3** Could the authors provide more details or justification, especially regarding how inter-layer dependencies are handled? The symmetry analysis of stacked layers is generally non-trivial, and additional explanation would strengthen the result.

**Q4** The experimental section is difficult to follow. Could the authors clarify how the proposed symmetry is applied in each experiment and how it influences optimization or the reported outcomes? Moreover, please provide more detailed information on the experimental settings, including hyperparameters and configuration details.

*Note*: The paper is difficult to follow for readers interested in both the theoretical and empirical aspects. The mathematical parts are presented with heavy formalism but without sufficient explanation or intuition, while the experimental section lacks important details and completeness. Addressing these issues would be important for me to properly evaluate the paper.

---

### Note · Authors · 2025-11-12

**Comment:**

I learned that an early version of the paper was accepted by a neurips workshop on 12/7/2025. So I will use that venue to share this work.

thanks to the reviewers for the great feedback

**Withdrawal Confirmation:**

I have read and agree with the venue's withdrawal policy on behalf of myself and my co-authors.